

# Matrix product operator symmetries and intertwiners in string-nets with domain walls

Laurens Lootens[1*], Jürgen Fuchs[2], Jutho Haegeman[1],
Christoph Schweigert[3] and Frank Verstraete[1]

**1** Department of Physics and Astronomy, Ghent University,
Krijgslaan 281, S9, B-9000 Ghent, Belgium
**2** Teoretisk fysik, Karlstads Universitet, Universitetsgatan 21, S-65188 Karlstad, Sweden
**3** Fachbereich Mathematik, Universität Hamburg, Bereich Algebra und Zahlentheorie,
Bundesstraße 55, D-20146 Hamburg, Germany

⋆ laurens.lootens@ugent.be

## Abstract

We provide a description of virtual non-local matrix product operator (MPO) symmetries in projected entangled pair state (PEPS) representations of string-net models. Given such a PEPS representation, we show that the consistency conditions of its MPO symmetries amount to a set of six coupled equations that can be identified with the pentagon equations of a bimodule category. This allows us to classify all equivalent PEPS representations and build MPO intertwiners between them, synthesising and generalising the wide variety of tensor network representations of topological phases. Furthermore, we use this generalisation to build explicit PEPS realisations of domain walls between different topological phases as constructed by Kitaev and Kong [Commun. Math. Phys. 313 (2012) 351-373]. While the prevailing abstract categorical approach is sufficient to describe the structure of topological phases, explicit tensor network representations are required to simulate these systems on a computer, such as needed for calculating thresholds of quantum error-correcting codes based on string-nets with boundaries. Finally, we show that all these string-net PEPS representations can be understood as specific instances of Turaev-Viro state-sum models of topological field theory on three-manifolds with a physical boundary, thereby putting these tensor network constructions on a mathematically rigorous footing.



# 1   Introduction

A large class of (2+1)D spin systems exhibiting topological order can be constructed using the string-net condensation mechanism introduced by Levin and Wen [1], leading to the string-net models which are classified by the data of a unitary fusion category (UFC) $\mathcal{D}$. Tensor network descriptions for string-net states have been obtained in [2–4] as PEPS wave functions. The fact that these PEPS wave functions exhibit non-trivial topological order can be related to local properties of the PEPS tensors in that they must exhibit non-trivial MPO symmetries [5]. The set of MPO symmetries is finite and closed under multiplication, and their properties can be described by the data of a UFC $\mathcal{C}$ [4,6]. Such MPO symmetries can then be used to construct the different ground states of the string-net model on a torus, as well as its anyonic excitations through an explicit construction of the Ocneanu tube algebra elements which form a basis for $Z(\mathcal{C})$, the monoidal center of $\mathcal{C}$. Recently, these constructions have also been related to subfactor theory concepts [7].

The main question we want to answer in this work is the following: given some string-net model $\mathcal{D}$, what are its possible PEPS representations and the corresponding MPO symmetries? We will show that the various consistency equations regarding the presence of MPO symmetries $\mathcal{C}$ in a given PEPS representation of a string-net $\mathcal{D}$ amount to the various pentagon equations in a $(\mathcal{C}, \mathcal{D})$-bimodule category $\mathcal{M}$, and that explicit representations of these tensors can be obtained using the associators of this bimodule category. In this way, we provide a framework that allows for a unified treatment and generalisation of known PEPS representations for quantum doubles, twisted quantum doubles and string-nets, and hence provides an important step into the realisation of a general "fundamental theorem" of PEPS [8].

Different PEPS representations of the same string-net model $\mathcal{D}$ should be locally indistinguishable, as the local physical properties are completely fixed upon specification of $\mathcal{D}$. We make this explicit by defining an MPO intertwiner that acts as an interface between two different PEPS representations, which can be pulled freely through the lattice and therefore has no effect on the local observables of the PEPS. Such an MPO intertwiner can be thought of as a generalisation of a virtual gauge transformation of the PEPS, and provides more insight into the precise nature of different PEPS representations that locally describe the same state.

The different torus ground states of these representations can be explicitly constructed using the MPO symmetries $\mathcal{C}$, which depends on the representation under consideration. Therefore, for these different MPO symmetries $\mathcal{C}_1, \mathcal{C}_2, \dots$ to describe the same ground state space, we need a 2-Morita equivalence between the fusion categories $\mathcal{C}_1, \mathcal{C}_2, \dots$ which can be guaranteed by requiring the bimodule category describing the relevant tensors to be invertible. We provide a new necessary condition on the associators of such a bimodule category as a reformulation of MPO-injectivity [5], a necessary requirement for describing topologically ordered systems with tensor networks.

Bimodule categories have previously been shown to be the relevant mathematical structure governing the properties of domain walls between two different string-net models [9], of which the case of a string-net on a manifold with boundaries is a special case. Using the more general PEPS representations for string-nets mentioned above, we define explicit tensor network representations for these boundaries and domain walls. While these features have been fully understood abstractly in the setting of category theory [9–13], tensor networks allow to devise actual tensors with all required properties and put those to work on the computer. This situation is similar to the difference between 6j symbols and 3j symbols in group theory: while 6j symbols are sufficient to think about the structure of representations, we need the Clebsch-Gordan coefficients to build effective models realizing those symmetries. Understanding boundaries and domain walls in this way is especially relevant in the context of topological quantum computation [14], since it is expected that it is much simpler to design physical systems with open boundary conditions rather than having to engineer the physical system in such a way that it effectively has the topology of a torus. Furthermore, these models show a much richer excitation spectrum in the presence of boundaries and domain walls, which in turn implies a larger possible gate set for such a topological quantum computer [15, 16].

We conclude with showing that these more general PEPS representations of string-net models, as well as the MPO symmetries and intertwiners can be understood in the framework of Turaev-Viro state-sum models. We show that the PEPS can be interpreted as a Turaev-Viro construction on a particular three-manifold, generalising a relation that has previously been reported in [17]. The benefit of framing these tensor network constructions in this way is twofold: on one hand, it allows one to prove properties of these tensor networks using TFT arguments, of which the relation between MPO-injectivity and Morita equivalence is an example. On the other hand we hope that this formulation will allow readers who are familiar with Turaev-Viro constructions to better understand these particular tensor networks and enrich the computational power for Turaev-Viro models with tensor network methods.

## 2   MPO symmetries

As mentioned in the introduction, topological order in a (2+1)D tensor network state requires the presence of virtual MPO symmetries that can be moved through the lattice. These MPO symmetries correspond to the Wegner-Wilson loop operators in string-net models, with the key difference that they act purely on the virtual level of the PEPS tensors rather than on the physical degrees of freedom. An important consequence of this is that these virtual MPO symmetries are unchanged when perturbing the state with some operator on the physical level, as opposed to the Wegner-Wilson loop operators which have to be dressed under such a perturbation [18]. In particular, these virtual symmetries also survive the dimensional reduction of the (2+1)D state to a 2D classical partition function by projecting out the physical degrees of freedom. Partition functions constructed in this way are expected to provide a description of critical lattice models governed by conformal field theories in their continuum limit [19, 20]. In this case, the MPO symmetries provide a lattice version of topological defects, which have been shown to encode symmetries and dualities of two-dimensional rational conformal field theories a long time ago [21]. The symmetries encoded by these defects commute with all chiral symmetries of the conformal field theory.

   In general, we expect those symmetries which are encoded in one-dimensional structures to be particularly robust, i.e. they are topological in nature and can be continuously deformed. In the situation considered here, the MPO symmetries can be pulled through the PEPS tensors which are located at the location of the physical degrees of freedom of the spin model. This motivates the *pulling-through condition*:

$$\begin{array}{c}\text{(diagram)}\end{array} \tag{1}$$

Here the oriented black lines represent the virtual legs of the PEPS tensor and thus the external legs of the MPO tensor, while the red lines are the internal legs of the MPO tensor (see Appendix A.3). Thus the MPO tensor itself is situated at the intersection between a red and black line, while the PEPS tensor is situated at the point at which three oriented black lines join and meet a physical leg (which is sticking out of the page), hence both the PEPS and MPO tensor are associated to *four*-valent vertices. This structural similarity will be heavily used later on. The pulling-through condition needs to be satisfied for each value of the physical index of the PEPS tensor. The Hilbert spaces associated to the physical index of the PEPS, to the virtual index of the PEPS and to the internal index of the MPO are respectively called $\mathcal{H}$, $\mathcal{V}$ and $\mathcal{W}$, and will be specified in Eqs. (15) – (17) below. For fixed $\mathcal{H}$ and $\mathcal{V}$, the space $\mathcal{W}$ will of course still depend upon the MPO under consideration.

   At first instance, such symmetry MPOs appear as closed loops and thus act as an operator on a tensor product of PEPS virtual indices. We jump back and forth between the closed MPOs and the individual MPO tensors. The closed objects are invariant under gauge transforms on the internal MPO index, while the pulling-through equation transforms covariantly under such a transformation. Such a gauge transform can be used to bring the MPO tensor in a block diagonal form, i.e. a direct sum of injective MPO tensors. At the level of the closed MPO, this implies that it is a (regular) sum of the operators associated with the injective MPO tensors, which themselves are also only defined up to a gauge transform. We now assume that there is a finite set of (isomorpism classes of) injective MPO tensors that satisfy Eq. (1), and any MPO tensor satisfying it can be decomposed as a direct sum thereof. We label (representatives of) these injective MPO tensors using $a$ and denote the corresponding closed MPO by the symbol $\hat{O}_a$.

   Clearly, the MPO tensor associated with the product of any two such MPOs also satisfies Eq. (1) and can thus be decomposed as a direct sum (at the level of the tensors) or a sum (at

the level of the operators) of our basis of injective MPOs. We can therefore restrict to work only with the basis of injective MPOs $\hat{O}_a$ and thereby obtain

$$\hat{O}_a \hat{O}_b = \sum_c N^c_{ab} \hat{O}_c \quad \Longleftrightarrow \quad \text{(diagram)} \quad (2)$$

with non-negative integers $N^c_{ab}$, implying that the set of MPOs form a fusion ring. [1] In order to be able to define an idempotent color, we promote this structure to an algebra over $\mathbb{C}$ by allowing for arbitrary complex linear combinations of the basis elements. In terms of the closed MPOs, this amounts to the presence of an additional virtual tensor, consisting of the expansion coefficients and commuting with the MPO tensor, in the trace in expression (69) that associates the operator $\hat{O}$ to an MPO tensor.

At the level of the MPO tensors, the decomposition (2) is established by the fusion tensors $X^{c,m}_{ab}$ that satisfy the zipper (i.e. intertwining) condition

$$\text{(diagram)} \quad (3)$$

where the tensor $X^{c,m}_{ab}$ is situated at the crossing of the three oriented red lines with $m = 1, 2, ..., N^c_{ab}$. The existence of these fusion tensors is guaranteed by the fundamental theorem of MPOs, which is a straightforward extension of the same theorem for MPS (see Appendix A for details); it states that two MPO tensors that generate the same MPO for any number of sites must be related by a gauge transformation on the internal MPO indices. This theorem is very straightforward to prove using the Cauchy-Schwarz inequality [22], but applied in this context plays a crucial role in translating the global multiplication property of these MPOs to local conditions on the MPO tensors. The multiplication of MPOs is associative, i.e. we have

$$(\hat{O}_a \hat{O}_b) \hat{O}_c = \hat{O}_a (\hat{O}_b \hat{O}_c). \quad (4)$$

At the level of the fusion tensors this imposes the recoupling identity [6]

$$\text{(diagram)} \quad (5)$$

where $^0F$ describes the basis transformation between the two fusion trees. This new quantity $^0F$ does not depend on the internal space. From this definition, it follows that it must satisfy a pentagon identity, which expresses the fact that the following two composite recoupling

---

[1] We may formalize the so obtained structure by considering a fusion category which has MPO tensors as objects and intertwiners between MPO tensors as morphisms. The MPOs are then associated with isomorphism classes of objects, and the injective MPOs with isomorphism classes of simple objects. The decomposition (2) realizes the tensor product of simple objects at the level of isomorphism classes.

procedures are equivalent:



Explicitly the pentagon identity for $^0F$ reads

$$\sum_o \left( {}^0F_e^{fcd}\right)_{g,lm}^{h,no}\left({}^0F_e^{abh}\right)_{f,ko}^{i,pq} = \sum_{j,rst}\left({}^0F_g^{abc}\right)_{f,kl}^{j,rs}\left({}^0F_e^{ajd}\right)_{g,sm}^{i,tq}\left({}^0F_i^{bcd}\right)_{j,rt}^{h,np}. \tag{6}$$

The simplest and first studied case of MPO symmetries are those where they are products of (unitary) group representations, i.e. the internal MPO space $\mathcal{W}$ is trivial (one-dimensional). The external MPO Hilbert space, which is the virtual PEPS space $\mathcal{V}$, can in that case be decomposed into a direct sum of irreducible representations of the group, on which the MPO tensors thus act block-diagonally, by using a suitable choice of basis of $\mathcal{V}$. Also in the general case it is useful to simultaneously block diagonalise all possible operators on $\mathcal{V}$ that appear in this set of MPOs, i.e. when interpreting the MPO tensors as operator-valued matrices, one should collect all operators appearing in those matrices for each of the injective MPO tensors $a$. These operators generate a subalgebra of $\mathrm{End}(\mathcal{V})$ that can be block diagonalised and gives a direct sum decomposition $\mathcal{V} \cong \bigoplus_\alpha \mathcal{V}_\alpha$, where we label the different blocks with Greek letters $\alpha, \beta, ...$

As by definition the MPOs act block-diagonally on the subspaces $\mathcal{V}_\alpha$, all of the above equations remain valid when restricting the external MPO index to the subspaces $\mathcal{V}_\alpha$. In particular, the pulling-through condition [Eq. (1)], which holds for arbitrary (direct) sums of the injective MPO tensors, can, for each value of the external index, be interpreted as a vertical intertwining relation between the horizontal concatenation of two MPO tensors (i.e. contracted along the internal MPO dimension) and a single MPO tensor. It thus makes sense to identify a basis of linearly independent fusion tensors $U_{\alpha\beta}^{\gamma,k}$ satisfying

$$\tag{7}$$

with $k = 1,...,N_{\alpha\beta}^\gamma$ a label for the linearly independent fusion tensors $U_{\alpha\beta}^{\gamma,k}$ situated at the crossing of the three oriented black lines, and $N_{\alpha\beta}^\gamma$ the dimension of this fusion space. This relations holds, with fixed fusion tensors $U_{\alpha\beta}^{\gamma,k}$, for every MPO, both for the set of injective MPO tensors as well as for any direct sum of them. As fusion of three concatenated MPO tensors must again be associative, we obtain an associativity condition for the fusion tensors $U_{\alpha\beta}^{\gamma,k}$

$$\sum_{\mu,mn} \left({}^4F_\delta^{\alpha\beta\gamma}\right)_{\mu,mn}^{\nu,jk} \tag{8}$$

with an associator $^4F$ that also satisfies a pentagon relation:

$$\sum_o \left(^4F_\rho^{\eta\gamma\delta}\right)_{\lambda,lm}^{\mu,no} \left(^4F_\rho^{\alpha\beta\mu}\right)_{\eta,ko}^{\nu,pq} = \sum_{\kappa,rst} \left(^4F_\lambda^{\alpha\beta\gamma}\right)_{\eta,kl}^{\kappa,rs} \left(^4F_\rho^{\alpha\kappa\delta}\right)_{\lambda,sm}^{\nu,tq} \left(^4F_\nu^{\beta\gamma\delta}\right)_{\kappa,rt}^{\mu,np}. \tag{9}$$

The set of labels $\alpha, \beta, \gamma, \ldots$ can in general be completely different from the labels $a, b, c, \ldots$; in particular, there is no need to identify $^4F$ with the associator $^0F$ we defined earlier.

In the most general case, identical blocks $\alpha$ may appear several times in the decomposition of $\mathcal{V}$, and denoting these degeneracies as $n_\alpha$, a more general way to write this decomposition is as $\mathcal{V} \cong \bigoplus_\alpha \bigoplus_{\mu_\alpha=1}^{n_\alpha} \mathcal{V}_\alpha \cong \bigoplus_\alpha \mathcal{V}_\alpha \otimes \mathbb{C}^{n_\alpha}$, where the operators in the MPO tensor thus act as the identity on the degeneracy spaces $\mathbb{C}^{n_\alpha}$. In order for the PEPS tensors to satisfy the pulling-through condition [Eq. (1)] for every value of the physical index, the PEPS tensor should itself be constructed as a linear map from the joint fusion spaces $\bigoplus_{\alpha,\beta,\gamma} \mathbb{C}^{N_{\alpha\beta}^\gamma} \otimes \mathbb{C}^{n_\alpha} \otimes \mathbb{C}^{n_\beta} \otimes \mathbb{C}^{n_\gamma}$ to the physical space $\mathcal{H}$. Put differently, for every value of the physical index, the PEPS tensor acts on the virtual space $\mathcal{V} \otimes \mathcal{V} \otimes \mathcal{V}$ as a linear combination (over $\alpha$, $\beta$, $\gamma$ and $k$) of the different fusion tensors $U_{\alpha,\beta}^{\gamma,k}$ multiplied with a arbitrary tensor on the degeneracy spaces $\mathbb{C}^{n_\alpha} \otimes \mathbb{C}^{n_\beta} \otimes \mathbb{C}^{n_\gamma}$. This is similar to how MPS or PEPS tensors with group symmetries are constructed [23,24], except that here the physical index is unaffected by the group action as the symmetry is purely virtual.

Henceforth we set $n_\alpha = 1$ and omit the corresponding tensor product factors, as the generalisation is straightforward. Furthermore, this minimal case is sufficient to understand the RG fixed point. Indeed, when the linear map that defines the PEPS tensor is isometric, which requires that

$$\mathcal{H} \cong \bigoplus_{\alpha,\beta,\gamma} \mathbb{C}^{N_{\alpha\beta}^\gamma}, \tag{10}$$

or at least that the physical space $\mathcal{H}$ contains the joint fusion spaces on the right hand side as a subspace, the associativity condition in Eq. (8) can be moved to the physical level, where it is one of the defining relation of the Levin-Wen string-net models. Ultimately, this associativity condition can be interpreted as a renormalization group (RG) transformation at the physical level, and thus expresses that the PEPS is an RG fixed point.

Contracting such an RG-invariant PEPS tensor and its conjugate along their physical index (which cancels the isometric linear map) then defines completely positive maps $T$ and $S$ that implement a fine- and coarse-graining of the corresponding MPOs

$$\tag{11}$$

and thus express scale invariance of the MPOs. In particular, this also holds for the projector MPO $\sum_a d_a O_a / D^2$, which is positive definite and thus represents a scale-invariant density matrix; here $D^2 = \sum_a d_a^2$, with the numbers $d_a$ being the quantum dimensions, as defined in Appendix B. The necessary existence of such completely positive maps for scale-invariant density matrices was proven in Ref. [25].

Henceforth, we identify $\mathcal{H} = \bigoplus_{\alpha,\beta,\gamma} \mathbb{C}^{N_{\alpha\beta}^\gamma}$ and just take the PEPS tensor to be equal to $U_{\alpha\beta}^{\gamma,k}$:

$$\tag{12}$$

whereby the physical index takes values in $(\alpha, \beta, \gamma, k)$. The structure of the vector spaces $\mathcal{V}$ and $\mathcal{W}$ (depending on specific injective MPOs of type $a$) is defined in the next section.

## 2.1 Bimodule categories

At this point, we have illustrated that MPO symmetries of PEPS define two fusion structures (more correct terminology: monoidal structures) corresponding to horizontal fusion of MPO products, and vertical fusion related to MPO scale transformations. They thus encode the algebraic data of two fusion categories $\mathcal{C}$ and $\mathcal{D}$. A particular prescription for such MPO tensors can be constructed by also invoking the algebraic data associated with a $(\mathcal{C}, \mathcal{D})$-bimodule category $\mathcal{M}$ (see appendix B)[2]. In particular, the pulling-through condition, the zipper condition, the two recoupling identities and the two pentagon equations coincide with the different pentagon equations of $\mathcal{C}$, $\mathcal{D}$ and $\mathcal{M}$ if we make the following identifications:

$$
\begin{aligned}
&:= \left(\frac{d_a d_b}{d_c}\right)^{\frac{1}{4}} \frac{\left({}^1F_A^{abC}\right)^{B,kj}_{c,mn}}{\sqrt{d_B}}, \qquad
&:= \left(\frac{d_a d_b}{d_c}\right)^{\frac{1}{4}} \frac{\left({}_1F_A^{abC}\right)^{B,kj}_{c,mn}}{\sqrt{d_B}}, \quad \text{(13a)}
\end{aligned}
$$

$$
\begin{aligned}
&:= \frac{\left({}^2F_B^{aC\alpha}\right)^{D,nk}_{A,jm}}{\sqrt{d_A d_D}}, \qquad
&:= \frac{\left({}_2F_B^{aC\alpha}\right)^{D,nk}_{A,jm}}{\sqrt{d_A d_D}}, \quad \text{(13b)}
\end{aligned}
$$

$$
\begin{aligned}
&:= \left(\frac{d_\alpha d_\beta}{d_\gamma}\right)^{\frac{1}{4}} \frac{\left({}^3F_B^{A\alpha\beta}\right)^{\gamma,km}_{C,jn}}{\sqrt{d_C}}, \qquad
&:= \left(\frac{d_\alpha d_\beta}{d_\gamma}\right)^{\frac{1}{4}} \frac{\left({}_3F_B^{A\alpha\beta}\right)^{\gamma,km}_{C,jn}}{\sqrt{d_C}}, \quad \text{(13c)}
\end{aligned}
$$

with

$$
\{a, b, c, \ldots\} \in I_{\mathcal{C}}, \quad \{A, B, C, \ldots\} \in I_{\mathcal{M}} \quad \text{and} \quad \{\alpha, \beta, \gamma, \ldots\} \in I_{\mathcal{D}},
$$

where $I_{\mathcal{C}}$, $I_{\mathcal{M}}$ and $I_{\mathcal{D}}$ are sets of representatives for isomorphism classes of simple objects in $\mathcal{C}$, $\mathcal{M}$ and $\mathcal{D}$ respectively. Here, ${}^1F$, ${}^2F$ and ${}^3F$ are the associators of $\mathcal{M}$ as a left $\mathcal{C}$-module category, a $(\mathcal{C}, \mathcal{D})$-bimodule category, and a right $\mathcal{D}$-module category, respectively; more details are provided in appendix B. The notation used to represent these tensors is a generalization of the triple line notation used in [2,3]; to interpret them in the usual tensor network sense it suffices to group the three lines and their multiplicity label into a single index, see appendix A.4. Using these definitions, the recoupling identity for the fusion tensors, the zipper equation, the pulling-through condition and the recoupling identity for the PEPS tensors are identified with pentagon equations $(P_1)$, $(P_2)$, $(P_3)$ and $(P_4)$ respectively (see Appendix B.4). Schematically, this identification amounts to

$$
{}^1F{}^1F = {}^0F{}^1F{}^1F \qquad {}^2F{}^1F = {}^1F{}^2F{}^2F \qquad {}^2F{}^3F = {}^3F{}^2F{}^2F \qquad {}^3F{}^3F = {}^4F{}^3F{}^3F
$$

---

[2]A bicategorical structure also controls full local rational conformal field theory [26, 27]. In [26], weak Hopf algebras are proposed as the underlying algebraic structure; this framework is more restrictive than the categorical setup we are using (and omits the pivotal structure on the module category). The notation we use for the fusion symbols of a two-object bicategory agrees with the notation in [26] and with the literature on weak Hopf algebras [28].

where we have omitted all labels and sums for simplicity. The quantum dimensions $d_a$, $d_A$, and $d_\alpha$ are positive real numbers associated to the simple objects of $\mathcal{C}$, $\mathcal{M}$ and $\mathcal{D}$, respectively[3]. We adopt the convention of [4], which involves a factor of a quantum dimension for every closed loop in a tensor network contraction. The presence of such factors can be avoided by locally absorbing these contributions into the definition of the relevant tensors, as was done in [6]; however, this requires prior knowledge of the lattice geometry, and for the sake of generality we do not use that approach. The quantum dimensions also appear in the definitions (13) because we not only want the pulling through property to hold for the specific case of equation (1), but also for more general cases such as

$$
\text{(14)}
$$

To achieve this, we must require that the fusion categories $\mathcal{C}$ and $\mathcal{D}$ have a spherical pivotal structure. One can then show that using the above definitions the left- and right-handed MPO tensors of (13b) are related by a gauge transformation on the internal legs, which ultimately allows one to prove more general cases of the pulling-through property. For details, we refer to appendix C. The vector spaces associated with the different indices can depend on orientation, but are of the general form

$$
\mathcal{V} = \bigoplus_{\alpha \in I_\mathcal{D}} \bigoplus_{A,B \in I_\mathcal{M}} \text{Hom}_\mathcal{M}(A \triangleleft \alpha, B),
\tag{15}
$$

for the virtual PEPS index, and

$$
\mathcal{W} = \bigoplus_{A,B \in I_\mathcal{M}} \text{Hom}_\mathcal{M}(a \triangleright A, B)
\tag{16}
$$

for the internal index of an injective MPO of type $a \in I_\mathcal{C}$, or with an additional direct sum over $a$ in a more general MPO. Here, $\triangleleft$ and $\triangleright$ denote the right and left action of $\mathcal{D}$ and $\mathcal{C}$, respectively, on the bimodule category $\mathcal{M}$. The physical space of the PEPS tensor is similarly rewritten as

$$
\mathcal{H} = \bigoplus_{\alpha,\beta,\gamma \in I_\mathcal{D}} \text{Hom}_\mathcal{D}(\alpha \otimes \beta, \gamma).
\tag{17}
$$

## 2.2   MPO intertwiners

Given a string-net model based on the fusion category $\mathcal{D}$, a PEPS representation can be determined by choosing a right $\mathcal{D}$-module category $\mathcal{M}$ and using the corresponding associator $^3F$ in definition (13c). Different choices of right $\mathcal{D}$-module categories $\mathcal{M}$ lead to different PEPS representations of the same string-net model $\mathcal{D}$. Because the underlying string-net model is the same, these PEPS wave functions are states in the same Hilbert space and can only differ globally; locally, they are indistinguishable. We can also choose different representations in different regions of the lattice; this situation is depicted in Figure 1a for two choices of right $\mathcal{D}$-module categories $\mathcal{M}_1$ and $\mathcal{M}_2$. The two PEPS representations, which we label $\text{PEPS}_{\mathcal{M}_1,\mathcal{D}}$ and $\text{PEPS}_{\mathcal{M}_2,\mathcal{D}}$, are separated by an *MPO intertwiner* (drawn in purple). The requirement that

---

[3] The existence of dimensions for objects in the fusion categories $\mathcal{C}$ and $\mathcal{D}$ requires them to be spherical fusion categories (see appendix C). A $(\mathcal{C}, \mathcal{D})$-bimodule category $\mathcal{M}$ does not come with an intrinsic duality, and therefore one can not a priori define dimensions for its objects. However, one can take the dual of an object $A \in \mathcal{M}$ to be the object $\bar{A}$ in the opposite category $\mathcal{M}^{\text{op}}$ where we identify $\text{Hom}_{\mathcal{M}^{\text{op}}}(\bar{A}, \bar{B})$ with $\text{Hom}_\mathcal{M}(B, A)$. $\mathcal{M}^{\text{op}}$ has the natural structure of a right $\mathcal{C}$-module and left $\mathcal{D}$-module category, which is unique because $\mathcal{C}$ and $\mathcal{D}$ are pivotal. Using $\mathcal{M}^{\text{op}}$, one can define dimensions for objects in $\mathcal{M}$; see e.g. [29].



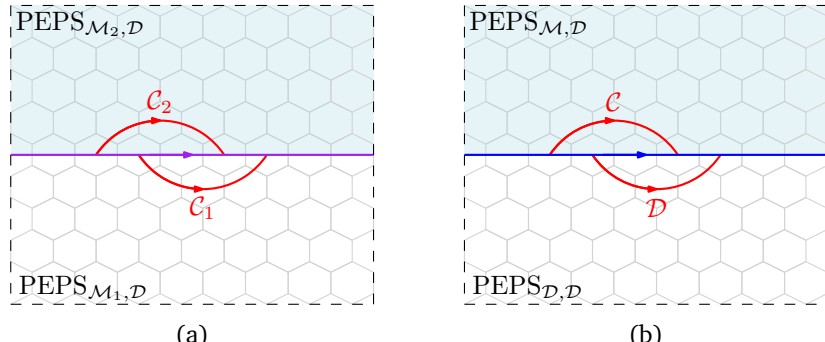

Figure 1: (a) Two PEPS representations for the ground state of the same string net model $\mathcal{D}$, determined by module categories $\mathcal{M}_1$ and $\mathcal{M}_2$. (b) For the particular case that $\mathcal{M}_1 = \mathcal{D}$ and $\mathcal{M}_2 = \mathcal{M}$, explicit tensors can be constructed in terms of the $F$-symbols of $\mathcal{M}$ as a $(\mathcal{C}, \mathcal{D})$-bimodule category.

these two representations should be locally indistinguishable then translates to the fact we should be able to move these MPO intertwiners freely through the lattice.

The choice of right $\mathcal{D}$-module category also determines the explicit representations of the MPO symmetry algebra $\mathcal{C}$. As argued above, in order for the MPO symmetries $\mathcal{C}$ to describe the excitations of the string-net $\mathcal{D}$ we should have a Morita equivalence between $\mathcal{C}$ and $\mathcal{D}$ which can be guaranteed by choosing $\mathcal{C} = \mathcal{D}^*_{\mathcal{M}}$, the dual of $\mathcal{D}$ with respect to $\mathcal{M}$. For the situation depicted in Figure 1a, we get MPO symmetries $\mathcal{C}_1$ and $\mathcal{C}_2$ described explicitly by tensors determined by the ${}^2F$ symbols of a $(\mathcal{C}_1, \mathcal{D})$-bimodule category $\mathcal{M}_1$ and $(\mathcal{C}_2, \mathcal{D})$-bimodule category $\mathcal{M}_2$, respectively. Since the product of an MPO symmetry with an MPO intertwiner is also an MPO intertwiner, the fundamental theorem again dictates that there should exist fusion tensors that decompose this product into a basis of injective MPO intertwiners. These fusion tensors can then be used to start and end MPO symmetries on MPO intertwiners.

For the specific case of $\mathcal{M}_1 = \mathcal{D}$ and $\mathcal{M}_2 = \mathcal{M}$ depicted in Figure 1b, we can find explicit representations of the MPO intertwiners (now drawn in blue) and corresponding fusion tensors. These tensors must satisfy various consistency conditions. First of all, the MPO intertwiners should be moveable through the lattice so as to guarantee that it is not locally observable:

$$ \vcenter{\hbox{$\begin{matrix} \alpha \searrow \quad \swarrow \beta \\ \boxed{k} \\ A \rightarrow \quad \uparrow \gamma \end{matrix}$}} = \vcenter{\hbox{$\begin{matrix} \alpha \quad \beta \\ \overset{\frown}{k} \\ A \uparrow \quad \gamma \end{matrix}$}} \quad . \tag{18} $$

The fusion tensors for multiplying an MPO intertwiner with an MPO symmetry satisfy

$$ \vcenter{\hbox{$\begin{matrix} \alpha \\ C \rightarrow m \quad a \\ B \end{matrix}$}} = \vcenter{\hbox{$\begin{matrix} \alpha \\ C \rightarrow m \quad a \\ B \end{matrix}$}} \tag{19} $$

for left multiplication and

$$ \vcenter{\hbox{$\begin{matrix} \alpha \\ C \rightarrow m \quad A \\ \beta \end{matrix}$}} = \vcenter{\hbox{$\begin{matrix} \alpha \\ C \rightarrow m \quad A \\ \beta \end{matrix}$}} \tag{20} $$

for right multiplication. Fusion of MPO symmetries with MPO intertwiners should be associa-

tive:

$$
\begin{array}{c} a \searrow \ \swarrow b \\ \overset{j}{\underset{c}{\bullet}} \\ A \rightarrow k \rightarrow B \end{array} = \sum_{C,mn} \left({}^{1}F_{A}^{abB}\right)_{c,jk}^{C,mn} \quad \begin{array}{c} a \quad b \\ \uparrow \quad \uparrow \\ A \rightarrow n \rightarrow m \rightarrow B \\ C \end{array} ,
$$ (21)

$$
\begin{array}{c} a \\ \uparrow \\ A \rightarrow k \underset{C}{\bullet} j \rightarrow B \\ \uparrow \\ \alpha \end{array} = \sum_{D,mn} \left({}^{2}F_{A}^{aB\alpha}\right)_{C,jk}^{D,mn} \quad \begin{array}{c} a \\ \uparrow \\ A \rightarrow n \underset{D}{\bullet} m \rightarrow B \\ \uparrow \\ \alpha \end{array} ,
$$ (22)

$$
\begin{array}{c} A \rightarrow k \underset{C}{\bullet} j \rightarrow B \\ \uparrow \quad \uparrow \\ \beta \quad \alpha \end{array} = \sum_{\gamma,mn} \left({}^{3}F_{A}^{B\alpha\beta}\right)_{C,jk}^{\gamma,mn} \quad \begin{array}{c} A \rightarrow n \rightarrow B \\ \overset{\gamma}{\underset{m}{\bullet}} \\ \beta \quad \alpha \end{array} .
$$ (23)

The associativity equations correspond to those of a bimodule category, showing that the injective MPO intertwiners can be labeled by simple objects in a $(\mathcal{C}, \mathcal{D})$-bimodule category $\mathcal{M}$, as already indicated by the notation. These consistency conditions can be satisfied by making the following identifications for the above tensors and their inverses:

$$
\begin{array}{c} A \underset{\gamma}{\bigg\vert} \overset{m}{\underset{C}{\bigg\vert}} B \\ j \longrightarrow C \longrightarrow k \\ \alpha \bigg\vert \underset{n}{\bigg\vert} \beta \end{array} := \frac{\left({}^{3}F_{B}^{C\alpha\gamma}\right)_{A,jm}^{\beta,nk}}{\sqrt{d_{A}d_{\beta}}}, \qquad \begin{array}{c} \alpha \overset{n}{\underset{\gamma}{\bigg\vert}} \beta \\ j \longleftarrow C \longrightarrow k \\ A \underset{m}{\bigg\vert} B \end{array} := \frac{\left({}_{3}F_{B}^{C\alpha\gamma}\right)_{A,jm}^{\beta,nk}}{\sqrt{d_{A}d_{\beta}}},
$$ (24a)

$$
\begin{array}{c} \overset{j}{\underset{a}{\bigg\uparrow}} \\ A \underset{C}{\bigg\langle} \underset{m}{\bullet} \underset{B}{\bigg\rangle} D \\ n \xrightarrow{C} m \xrightarrow{B} k \\ \alpha \end{array} := \left(\frac{d_{a}d_{B}}{d_{C}}\right)^{\frac{1}{4}} \frac{\left({}^{2}F_{A}^{aB\alpha}\right)_{C,mn}^{D,kj}}{\sqrt{d_{D}}}, \qquad \begin{array}{c} \overset{j}{\underset{a}{\bigg\downarrow}} \\ D \underset{m}{\bigg\langle} \underset{C}{\bigg\rangle} A \\ k \xrightarrow{B} m \xrightarrow{C} n \\ \alpha \end{array} := \left(\frac{d_{a}d_{B}}{d_{C}}\right)^{\frac{1}{4}} \frac{\left({}_{2}F_{A}^{aB\alpha}\right)_{C,mn}^{D,kj}}{\sqrt{d_{D}}}, \quad (24b)
$$

$$
\begin{array}{c} B \\ n \xrightarrow{C} m \xrightarrow{A} j \\ \alpha \underset{\beta}{\bigg\rangle} \underset{k}{\bigg\vert} \gamma \end{array} := \left(\frac{d_{A}d_{\beta}}{d_{C}}\right)^{\frac{1}{4}} \frac{\left({}^{3}F_{B}^{A\beta\alpha}\right)_{C,mn}^{\gamma,kj}}{\sqrt{d_{\gamma}}}, \qquad \begin{array}{c} B \\ j \xrightarrow{A} m \xrightarrow{C} n \\ \gamma \underset{\beta}{\bigg\vert} \underset{k}{\bigg\rangle} \alpha \end{array} := \left(\frac{d_{A}d_{\beta}}{d_{C}}\right)^{\frac{1}{4}} \frac{\left({}_{3}F_{B}^{A\beta\alpha}\right)_{C,mn}^{\gamma,kj}}{\sqrt{d_{\gamma}}}, \quad (24c)
$$

such that the consistency conditions coincide with the following pentagon equations in a $(\mathcal{C}, \mathcal{D})$-bimodule category $\mathcal{M}$:

$$
{}^{3}F{}^{3}F = {}^{3}F{}^{3}F{}^{4}F , \qquad {}^{3}F{}^{2}F = {}^{2}F{}^{2}F{}^{3}F , \qquad {}^{3}F{}^{3}F = {}^{3}F{}^{3}F{}^{4}F ,
$$

$$
{}^{1}F{}^{2}F = {}^{1}F{}^{2}F{}^{2}F , \qquad {}^{3}F{}^{2}F = {}^{2}F{}^{2}F{}^{3}F , \qquad {}^{3}F{}^{3}F = {}^{3}F{}^{3}F{}^{4}F .
$$

The more general case of Figure 1*a* requires more general $F$ symbols than what we can get from a bimodule category; we refer to Section 5 for more details. We stress that these different PEPS

representations of the same string-net model $\mathcal{D}$ are only locally indistinguishable. Indeed, if we define these PEPS on a manifold with a non-trivial topology these different representations may represent different ground states. To see this explicitly, consider two PEPS representations $|\psi_1\rangle$ and $|\psi_2\rangle$ of the string-net model $\mathcal{D}$ on a torus:

$$|\psi_1\rangle = \boxed{\text{PEPS}_{\mathcal{D},\mathcal{D}}}, \quad |\psi_2\rangle = \boxed{\text{PEPS}_{\mathcal{M},\mathcal{D}}}. \tag{25}$$

We can insert a small region of a different representation $\text{PEPS}_{\mathcal{M},\mathcal{D}}$ in the state $|\psi_1\rangle$ by inserting a closed loop of MPO intertwiner labeled by $A$ at the cost of multiplying with the quantum dimension $d_A$ associated to this MPO intertwiner. We can now grow this closed loop so as to cover almost the entire torus by $\text{PEPS}_{\mathcal{M},\mathcal{D}}$, and then reduce the so obtained loop to a network of MPOs whose edges are labeled by simple objects in $\mathcal{C}$:

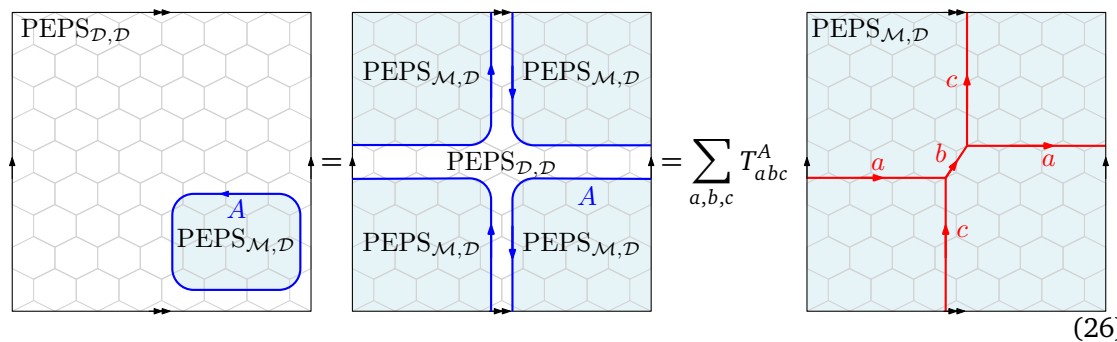

$$\tag{26}$$

where $T^A_{abc}$ are coefficients for the tube algebra elements built using MPOs in $\mathcal{C}$. This means that, unless all MPOs $a, b, c$ in the final result are labeled by the identity object, $|\psi_1\rangle$ and $|\psi_2\rangle$ are not the same state. Generically, this is indeed what happens; we will illustrate this with a simple example using the toric code in Section 4. An important observation here is that even though $T^A_{abc}$ depends on the initial choice of MPO intertwiner $A$, the resulting ground state is the same for all these choices up to a factor. This is related to the fact that the torus ground states are characterised by the central idempotents of the tube algebra, which can be non-uniquely decomposed into simple idempotents that each represent the same ground state, an observation made in [6]. A more detailed explanation requires the explicit computation of $T^A_{abc}$, which we leave for future work.

## 2.3 2-Morita equivalence and MPO injectivity

The ground states on a torus in a string-net model $\mathcal{D}$ are well understood. They are described by the monoidal center $Z(\mathcal{D})$ [1]. At the same time, these ground states can be understood as idempotents of the tube algebra [4,6] associated to the MPO symmetries $\mathcal{C}$, which is an explicit construction of the monoidal center $Z(\mathcal{C})$. Therefore, in order for the MPO symmetries $\mathcal{C}$ to correctly describe the ground states of some string-net model $\mathcal{D}$, we need the monoidal centers of these two fusion categories to be equivalent, $Z(\mathcal{D}) \simeq Z(\mathcal{C})$. Fusion categories $\mathcal{D}$ and $\mathcal{C}$ with this property are said to be *2-Morita equivalent*. It can be shown that $\mathcal{D}$ and $\mathcal{C}$ are 2-Morita equivalent if and only if there exists a $(\mathcal{C}, \mathcal{D})$-bimodule category $\mathcal{M}$ satisfying

$$\mathcal{M} \boxtimes_{\mathcal{D}} \mathcal{M}^{\text{op}} \simeq \mathcal{C} \quad \text{and} \quad \mathcal{M}^{\text{op}} \boxtimes_{\mathcal{C}} \mathcal{M} \simeq \mathcal{D}, \tag{27}$$

with $\boxtimes_{\mathcal{C}}$ and $\boxtimes_{\mathcal{D}}$ denoting the Deligne product relative to $\mathcal{C}$ and $\mathcal{D}$ respectively [4]. Such a bimodule category $\mathcal{M}$ is called an *invertible* $(\mathcal{C}, \mathcal{D})$-bimodule category [32]. If we use the associators of this invertible bimodule category to define the PEPS, MPO and fusion tensors, the MPO symmetries $\mathcal{C}$ are guaranteed to correctly describe the ground state subspace of the string-net $\mathcal{D}$. These relations can be interpreted as the requirement that two PEPS representations $\text{PEPS}_{\mathcal{M}, \mathcal{D}}$ and $\text{PEPS}_{\mathcal{D}, \mathcal{D}}$ describe the same ground state manifold, as it ensures that there is an invertible isomorphism between the two tube algebras as constructed from their respective MPO symmetries $\mathcal{C}$ and $\mathcal{D}$.

The standard definition of an invertible bimodule category in terms of the equivalences (27) is not practical for explicit computations. For our purposes, we would rather like to have a condition on the associators of a given bimodule category that allows one to decide whether or not it is invertible. This is a well posed problem, as the bimodule category is completely fixed upon specifying its associators; to our knowledge it has not yet been assessed in the literature.

In [5], a condition relating the MPO symmetries and the PEPS tensors was derived that ensures that the ground state degeneracy of the PEPS on a particular manifold does not depend on the system size, a necessary requirement for a topologically ordered state at an RG fixed point. This condition, which is known as *MPO-injectivity*, requires that the PEPS tensor, when viewed as a map from its virtual spaces to the physical space, is invertible on a subspace provided by some MPO. Applied to the case under consideration in this paper, this condition can be written as

$$
\vcenter{\hbox{\includegraphics{lhs}}} = \sum_a \frac{d_a}{D^2} \vcenter{\hbox{\includegraphics{rhs}}} , \tag{28}
$$

where the physical indices of the PEPS and its pseudoinverse are contracted and we have numbered the legs of the PEPS and MPO tensors to indicate the explicit identification between the left- and right-hand side. Previously, this equation was shown to hold for the case where we define the PEPS and MPO tensors using $\mathcal{D}$ as a $(\mathcal{D}, \mathcal{D})$-bimodule category, as in that case it coincides with the pentagon equation. Using the associators of a generic $(\mathcal{C}, \mathcal{D})$-bimodule category $\mathcal{M}$ however, this equation can no longer be identified with any of its pentagon equations, and it is in fact straightforward to describe bimodule categories for which this equation does not hold.

One can show that the equality (28) holds whenever the $(\mathcal{C}, \mathcal{D})$-bimodule category $\mathcal{M}$ is invertible, meaning that the MPO-injectivity condition is a necessary requirement on the associators of a bimodule category for it to be invertible. Whether or not this is also a sufficient condition is left as an open question.

## 3  Boundaries and domain walls

The generalised PEPS representations discussed above now also allow us to describe domain walls between different string-net models $\mathcal{D}_1$ and $\mathcal{D}_2$. Following Kitaev and Kong [9], domain walls between two such models can be constructed using a $(\mathcal{D}_1, \mathcal{D}_2)$-bimodule category $\mathcal{M}$. These domain walls satisfy a set of consistency equations which can be thought of as generalisations of the Levin-Wen string-net condition in the form of Eq. (8) in the presence of such a

---

[4]The relative Deligne product can be understood at the level of simple objects as the Karoubi envelope of a structure known as the ladder category, i.e. $\mathcal{M}^{\text{op}} \boxtimes_{\mathcal{D}} \mathcal{M} \simeq \mathbf{Kar}(\mathbf{Lad}_{\mathcal{D}}(\mathcal{M}^{\text{op}}, \mathcal{M}))$. This formulation of the Deligne product can be worked out in the diagrammatic language following [30, 31] and is particularly suited for tensor network calculations.

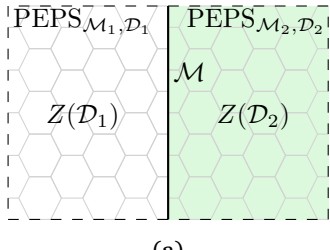
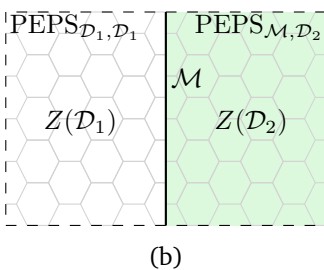

(a)                                             (b)

Figure 2: (a) A domain wall between two string-net models $\mathcal{D}_1$ and $\mathcal{D}_2$, described by $\text{PEPS}_{\mathcal{M}_1,\mathcal{D}_1}$ and $\text{PEPS}_{\mathcal{M}_2,\mathcal{D}_2}$ respectively. (b) For the particular case that $\mathcal{M}_1 = \mathcal{D}_1$ and $\mathcal{M}_2 = \mathcal{M}$, explicit tensors can be constructed in terms of $F$-symbols of a $(\mathcal{D}_1, \mathcal{D}_2)$-bimodule category $\mathcal{M}$.

domain wall. A priori, we can consider any PEPS representation of the string-nets $\mathcal{D}_1$ and $\mathcal{D}_2$, which is the situation depicted in Figure 2a. By restricting to the more specific case of Figure 2b however, we will be able to find explicit PEPS representations of the relevant domain wall tensors using the associators of a $(\mathcal{D}_1, \mathcal{D}_2)$-bimodule category $\mathcal{M}$. The general case of Figure 2a requires a further generalisation of these associators, which we briefly discuss at the end of Section 5.

Starting on the left side of the domain wall we have a string-net model $\mathcal{D}_1$ with the usual bulk Levin-Wen recoupling condition:

$$
\begin{array}{c}
\beta_1 \quad \gamma_1 \\
j \\
\nu_1 \\
k \\
\alpha_1 \quad \delta_1
\end{array}
= \sum_{\mu_1, mn} \left( {}^0 F^{\alpha_1 \beta_1 \gamma_1}_{\delta_1} \right)^{\nu_1, jk}_{\mu_1, mn}
\begin{array}{c}
\beta_1 \quad \gamma_1 \\
m \, \mu_1 \, n \\
\alpha_1 \quad \delta_1
\end{array},
\tag{29}
$$

with $\alpha_1, \beta_1, \dots$ labelling simple objects in $\mathcal{D}_1$. Boundaries for this model are described by module categories $\mathcal{M}$ over $\mathcal{D}_1$ with the associativity condition

$$
\begin{array}{c}
B \\
\beta_1 \to j \\
\alpha_1 \to k \\ C \\
A
\end{array}
= \sum_{\gamma_1, mn} \left( {}^1 F^{\alpha_1 \beta_1 B}_A \right)^{C, jk}_{\gamma_1, mn}
\begin{array}{c}
\beta_1 \quad B \\
m \, \gamma_1 \, n \\
\alpha_1 \\
A
\end{array}.
\tag{30}
$$

Here, $\mathcal{M}$ is a left $\mathcal{D}_1$-module category with simple objects labeled by $A, B, \dots$. A domain wall can be described by having $\mathcal{M}$ also be a boundary for a string-net model $\mathcal{D}_2$ with simple objects $\alpha_2, \beta_2, \dots$. This boundary is a right $\mathcal{D}_2$-module category, which requires the following associativity conditions:

$$
\begin{array}{c}
B \\
k \, \gamma_2 \, j \quad \alpha_2 \\
A \qquad \beta_2
\end{array}
= \sum_{C, mn} \left( {}^3 F^{B \alpha_2 \beta_2}_A \right)^{\gamma_2, jk}_{C, mn}
\begin{array}{c}
B \\
m \to \alpha_2 \\
C \\
n \to \beta_2 \\
A
\end{array}
\tag{31}
$$

$$
\begin{array}{c}
\alpha_2 \qquad \beta_2 \\
k \, \nu_2 \, j \\
\delta_2 \qquad \gamma_2
\end{array}
= \sum_{\mu_2, mn} \left( {}^4 F^{\alpha_2 \beta_2 \gamma_2}_{\delta_2} \right)^{\nu_2, jk}_{\mu_2, mn}
\begin{array}{c}
\alpha_2 \quad \beta_2 \\
m \\
\mu_2 \\
n \\
\delta_2 \quad \gamma_2
\end{array}
\tag{32}
$$

The requirement that $\mathcal{M}$ is a domain wall then becomes

$$
\alpha_1 - k \underset{A}{\overset{B}{\underset{D}{\bigg|}}} \alpha_2 = \sum_{C,mn} \left(^2F_A^{\alpha_1 B \alpha_2}\right)_{C,mn}^{D,jk} \quad \alpha_1 - \underset{A}{\overset{B}{\underset{n}{\bigg|}}} C \alpha_2 \qquad . \tag{33}
$$

A tensor network representation of these domain wall structures for the specific case in Figure 2b is then given by the following assignments:

$$
\text{(34a)}
$$

$$
\text{(34b)}
$$

$$
\text{(34c)}
$$

$$
\text{(34d)}
$$

such that equations (29)-(33) become the following bimodule pentagon equations in a $(\mathcal{D}_1, \mathcal{D}_2)$-bimodule category $\mathcal{M}$:

$$
{}^0F{}^0F = {}^0F{}^0F{}^0F \qquad {}^1F{}^1F = {}^1F{}^0F{}^1F \qquad {}^2F{}^1F = {}^2F{}^1F{}^2F ,
$$

$$
{}^2F{}^3F = {}^3F{}^2F{}^2F \qquad\qquad {}^3F{}^3F = {}^4F{}^3F{}^3F .
$$

# 4 Examples

In this section we explicitly construct the various tensors discussed above for specific instances of the categories $\mathcal{D}$, $\mathcal{M}$ and $\mathcal{C}$. An explanation of the relevant bimodule categories can be found in Appendix D.

## 4.1 The toric code

The simplest and most well-known example of a system exhibiting topological order is the toric code, which is a $\mathbb{Z}_2$ quantum double model. We denote the elements of $\mathbb{Z}_2$ as $g \in \{+1, -1\}$. We will consider the fusion category $\text{Vec}_{\mathbb{Z}_2}$ of $\mathbb{Z}_2$-graded vector spaces. The simple objects of this category are one-dimensional $\mathbb{Z}_2$-graded vector spaces which we can label by the group elements of $\mathbb{Z}_2$, and the tensor product is given by the group multiplication. Since $\mathbb{Z}_2$ is an abelian group, its irreducible representations are 1-dimensional and its representation category $\text{Rep}(\mathbb{Z}_2)$ is monoidally equivalent to $\text{Vec}_{\mathbb{Z}_2}$.

$\underline{\mathcal{D} = \mathcal{M} = \mathcal{C} = \textbf{Vec}_{\mathbb{Z}_2}}$

The first representation of the toric code PEPS, MPO and fusion tensors can be obtained by choosing $\mathcal{D}$ as a module category over itself, and correspondingly also taking $\mathcal{C} = \mathcal{M} = \mathcal{D}$. This representation is the one studied in [4,6] and has the following explicit tensors:

$$\includegraphics = 1, \qquad \includegraphics = 1, \qquad \includegraphics = 1, \tag{35}$$

where here and henceforth $g_{12} := g_1 g_2$ and as usual we only define non-zero tensor entries. As all fusion spaces are 1-dimensional we simply denote the multiplicity label by 1. In this representation, the non-trivial MPO symmetry labeled by $g_1 = -1$ is a tensor product of local operators that act as Pauli $\sigma_x$ operators on the virtual loops of the PEPS representation.

$\underline{\mathcal{D} = \textbf{Vec}_{\mathbb{Z}_2}, \mathcal{M} = \textbf{Vec}, \mathcal{C} = \textbf{Rep}(\mathbb{Z}_2) \simeq \textbf{Vec}_{\mathbb{Z}_2}}$

A second representation of the toric code PEPS, MPO and fusion tensors can be obtained by choosing $\mathcal{M} = \text{Vec}$, i.e. the category of finite dimensional vector spaces. This category has only one simple object and lines labeled by this object will be drawn as dotted lines. We obtain an invertible $(\mathcal{C}, \mathcal{D})$-bimodule category $\mathcal{M}$ by choosing $\mathcal{C} = \text{Rep}(\mathbb{Z}_2) \simeq \text{Vec}_{\mathbb{Z}_2}$. This representation is the one used in [33] and has the following explicit tensors:

$$\includegraphics = 1, \qquad \includegraphics = \begin{cases} 1, & g_2 = +1, \\ g_1, & g_2 = -1, \end{cases} \qquad \includegraphics = 1. \tag{36}$$

In this representation, the non-trivial MPO symmetry labeled by $g_1 = -1$ is a tensor product of local operator that act as Pauli $\sigma_z$ operators on the physical degrees of freedom of the PEPS representation.

### 4.1.1 MPO intertwiner between different PEPS representations

On the physical level, the two representations $\mathrm{PEPS}_{\mathrm{Vec}_{\mathbb{Z}_2},\mathrm{Vec}_{\mathbb{Z}_2}}$ and $\mathrm{PEPS}_{\mathrm{Vec},\mathrm{Vec}_{\mathbb{Z}_2}}$ are locally indistinguishable. This property can be made very explicit by the existence of the following MPO intertwiner between these two representations:

$$
\text{(37)}
$$

along with corresponding fusion tensors to allow MPO symmetries in the two representations to start and end on an MPO intertwiner. The fact that these states are only locally the same state and globally might represent different ground states on a torus is exemplified by the following equality:

$$
\mathrm{PEPS}_{\mathrm{Vec}_{\mathbb{Z}_2},\mathrm{Vec}_{\mathbb{Z}_2}} = \sum_{g_1,g_2} \mathrm{PEPS}_{\mathrm{Vec},\mathrm{Vec}_{\mathbb{Z}_2}}, \tag{38}
$$

which can be obtained by a straightforward computation explained in Section 2.2. This implies that without the presence of MPO symmetries, the two PEPS representations represent different ground states on the torus.

### 4.1.2 Smooth and rough boundaries

It has long been known that the toric code admits two different types of boundary conditions if we consider it on a manifold with boundaries [34]. These two boundary conditions are called the smooth and rough boundary, and they differ in the effect they have on bulk excitations approaching these boundaries. They can be understood in the general framework of string-net boundaries as being represented by a domain wall to the vacuum, where the boundary labels are given by a $(\mathrm{Vec}_{\mathbb{Z}_2},\mathrm{Vec})$-bimodule category $\mathcal{M}$. The two choices for such a bimodule category are given by $\mathcal{M} = \mathrm{Vec}_{\mathbb{Z}_2}$ and $\mathcal{M} = \mathrm{Vec}$ describing the smooth and rough boundaries respectively. The PEPS representation of these smooth and rough boundary conditions is then respectively given by

$$
g_{123} = 1, \qquad \text{(39)} \qquad\qquad = 1. \qquad \text{(40)}
$$

Using these boundary conditions, one can explicitly compute what happens when excitations approach a boundary, and for the case of the toric code it turns out that electric excitations can condense on a smooth boundary, while magnetic excitations can condense on a rough boundary [35]. This can be studied using the construction of the tube algebra using MPO symmetries; we will explore this in detail in future work.

## 4.2   The $S_3$ quantum double

As a second example, let us consider the smallest non-abelian group, i.e. the permutation group of three elements $S_3$. We will consider the fusion category $\mathbf{Vec}_{S_3}$ of $S_3$-graded vector spaces. The simple objects of this category are one-dimensional $S_3$-graded vector spaces which we can label by the group elements of $S_3$, and the tensor product is given by the group multiplication. $S_3$ has three irreducible representations $\alpha \in \{0, \psi, \pi\}$ respectively corresponding to the trivial, sign and two-dimensional irrep. We can consider the representation category $\mathrm{Rep}(S_3)$ as a fusion category with simple objects labeled by irreps and the tensor product given by the fusion rules of these irreps. The quantum dimensions of these simple objects correspond to the dimensions of the irreps, i.e. $d_0 = d_\psi = 1, d_\pi = 2$. The non-trivial fusion rules in this case are given by

$$\psi \otimes \psi = 0, \quad \pi \otimes \psi = \psi \otimes \pi = \pi, \quad \pi \otimes \pi = 0 + \psi + \pi. \tag{41}$$

$\underline{\mathcal{D} = \mathcal{M} = \mathcal{C} = \mathbf{Vec}_{S_3}}$

Similar to the toric code, a representation of the $S_3$ quantum double PEPS, MPO and fusion tensors can be obtained by choosing $\mathcal{D}$ as a module category over itself, and correspondingly also taking $\mathcal{C} = \mathcal{M} = \mathcal{D}$. This representation is again the one studied in [4, 6] and has the following explicit tensors:

$$\tag{42}$$

where the MPOs labeled by $g$ now act as the left-regular representation $L_g$ on the virtual loops of the PEPS representation.

$\underline{\mathcal{D} = \mathbf{Vec}_{S_3}, \mathcal{M} = \mathbf{Vec}, \mathcal{C} = \mathbf{Rep}(S_3)}$

A second representation of the $S_3$ quantum double PEPS, MPO and fusion tensors can be obtained by choosing $\mathcal{M} = \mathrm{Vec}$. We obtain an invertible $(\mathcal{C}, \mathcal{D})$-bimodule category $\mathcal{M}$ by choosing $\mathcal{C} = \mathrm{Rep}(S_3)$. We obtain the following tensors:

$$\cdots = 1, \qquad j \xrightarrow[\ \alpha\ ]{g} k = D^\alpha(g)^k_j, \qquad \cdots = \left(\frac{d_{\alpha_1} d_{\alpha_2}}{d_{\alpha_3}}\right)^{\frac{1}{4}} C^{\alpha_1 \alpha_2 \alpha_3}_{i_1 i_2 i_3}.$$

$$\tag{43}$$

This is the first example we encounter where the fusion spaces are not simply one-dimensional, which is due to the fact that $S_3$ has a two-dimensional irreducible representation $\pi$. The MPOs are labeled by these irreducible representations, and the multiplication of two MPOs is simply a tensor product since the external MPO index is one-dimensional. The fusion tensors then are the Clebsch-Gordan coefficients intertwining the tensor product of two irreducible representations.

$\underline{\mathcal{D} = \mathcal{M} = \mathcal{C} = \textbf{Rep}(S_3)}$

We now study a different string-net model by taking $\mathcal{D} = \text{Rep}(S_3)$. By also choosing $\mathcal{M} = \mathcal{C} = \text{Rep}(S_3)$, we obtain the following tensors:

$$
\begin{array}{ccc} \text{(diagram)} & = & \left( \dfrac{d_{\alpha_2} d_{\alpha_3}}{d_{\alpha_6}} \right)^{\frac{1}{4}} \dfrac{\left( F_{\alpha_4}^{\alpha_1 \alpha_2 \alpha_3} \right)_{\alpha_5}^{\alpha_6}}{\sqrt{d_{\alpha_5}}}, \end{array}
$$

(44)

$$
\text{(diagram)} = \dfrac{\left( F_{\alpha_4}^{\alpha_1 \alpha_2 \alpha_3} \right)_{\alpha_5}^{\alpha_6}}{\sqrt{d_{\alpha_5} d_{\alpha_6}}}, \qquad \text{(diagram)} = \left( \dfrac{d_{\alpha_1} d_{\alpha_2}}{d_{\alpha_5}} \right)^{\frac{1}{4}} \dfrac{\left( F_{\alpha_4}^{\alpha_1 \alpha_2 \alpha_3} \right)_{\alpha_5}^{\alpha_6}}{\sqrt{d_{\alpha_6}}}.
$$

This is the representation studied in [6]. We note that the MPO symmetries again form a representation of $\text{Rep}(S_3)$, and therefore that the anyonic excitations of this PEPS representation must be the same as the previous two PEPS representations. This is due to the fact that the monoidal centers $Z(\text{Vec}_{S_3})$ and $Z(\text{Rep}(S_3))$ are isomorphic, which is true for any group $G$ and its representations $\text{Rep}(G)$.

$\underline{\mathcal{D} = \textbf{Rep}(S_3), \mathcal{M} = \textbf{Vec}, \mathcal{C} = \textbf{Vec}_{S_3}}$

The above string-net model admits another PEPS representation that can be obtained by choosing $\mathcal{M} = \text{Vec}$ and $\mathcal{C} = \text{Vec}_{S_3}$, giving the following tensors:

$$
\text{(diagram)} = \left( \dfrac{d_{\alpha_1} d_{\alpha_2}}{d_{\alpha_3}} \right)^{\frac{1}{4}} C_{i_1 i_2 i_3}^{\alpha_1 \alpha_2 \alpha_3}, \qquad \text{(diagram)} = D^{\alpha}(g)_j^k, \qquad \text{(diagram)} = 1.
$$

(45)

This representation is equivalent to the one in [33], since the left-regular representation is just the sum of all irreducible representations weighed by their dimension.

## 5 Turaev-Viro TFT, PEPS and MPO symmetries

The assignments made for the tensors in Equations (13) and (24) have been chosen in such a way that the various consistency conditions on those tensors all amount to some pentagon equation for a pair of fusion categories $\mathcal{C}$ and $\mathcal{D}$ together with a $(\mathcal{C}, \mathcal{D})$-bimodule category $\mathcal{M}$. In this section we explain how the specific form of these tensors can be derived from a Turaev-Viro state-sum construction of a 3d TFT. More precisely, the PEPS tensor network can be shown to be an instance of such a 3d Turaev-Viro TFT on a particular three-manifold with a choice of skeleton. Such a construction has already been performed in [17] for the particular PEPS representation that is discussed in [2,3], which corresponds to the case that $\mathcal{C} = \mathcal{D}$ with $\mathcal{M}$ being $\mathcal{D}$ as a bimodule category over itself. We present the construction for the general case of $(\mathcal{C}, \mathcal{D})$-bimodule categories considered in this paper, freely using the formulation of Turaev-Viro TFT given in [36] and extending it to the presence of a physical boundary.

Let us first restate what the PEPS tensor network really represents. We consider an oriented surface $\Sigma$ with a cell decomposition $\Delta$. For concreteness we take $\Delta$ to be the honeycomb lattice, but the construction works analogously for arbitrary cell decompositions including cases where the vertices of the cell decomposition do not all have the same number of legs. We recall from Section 2 that the Hilbert space $\mathcal{H}$ associated to the physical leg of the PEPS tensor (13c) is given by the direct sum

$$\mathcal{H} = \bigoplus_{\alpha,\beta,\gamma \in I_{\mathcal{D}}} \mathrm{Hom}_{\mathcal{D}}(\alpha \otimes \beta, \gamma). \tag{46}$$

Denoting by $\Delta_0$ the set of vertices of the cell decomposition, we can associate a Hilbert space $\mathcal{H}_\Sigma$ to the surface $\Sigma$ and its decomposition $\Delta$ by attaching a copy of $\mathcal{H}$ to every vertex:

$$\mathcal{H}_\Sigma = \bigotimes_{v \in \Delta_0} \mathcal{H}. \tag{47}$$

Placing a trivalent PEPS tensor at each of the vertices and contracting along the edges of $\Delta$, the PEPS tensor network describes a state in a "space of ground states" or "protected space" $\mathcal{H}_\Sigma^0 \subseteq \mathcal{H}_\Sigma$, the dimension of which depends on the topology of the surface $\Sigma$.

We are now going to show that the subspace $\mathcal{H}_\Sigma^0$ is also provided by a Turaev-Viro construction, and that we recover the PEPS representation (13c) from that construction. To do so, we introduce the three-manifold

$$M_\Sigma := \Sigma \times [0, 1]. \tag{48}$$

The geometric boundary $\partial M_\Sigma$ of $M_\Sigma$ is the disjoint union of two copies $\Sigma \times \{0\}$ and $\Sigma \times \{1\}$ of the surface $\Sigma$. We regard $\Sigma \times \{1\}$ as a *gluing boundary*. This is the type of boundary that arises when chopping a three-manifold into pieces; to such a boundary the Turaev-Viro TFT for $\mathcal{D}$, to be denoted as $\mathrm{TFT}_{\mathcal{D}}$, assigns the vector space

$$\mathrm{TFT}_{\mathcal{D}}(\Sigma) = \mathcal{H}_\Sigma^0. \tag{49}$$

In contrast, the other part $\Sigma \times \{0\}$ of $\partial M_\Sigma$ is taken to be a *physical* or *end-of-the-world* boundary (sometimes also called a *brane* boundary in the literature). On a physical boundary a boundary condition must be specified. In the case of $\mathrm{TFT}_{\mathcal{D}}$ the elementary boundary conditions are labeled by indecomposable right $\mathcal{D}$-module categories [9, 37], which we will denote by $\mathcal{M}$. Note that $\mathcal{M}$ has a natural structure of a left module category over the "dual" of $\mathcal{D}$ with respect to $\mathcal{M}$, i.e. over the monoidal category $\mathcal{C} = \mathcal{D}_\mathcal{M}^*$ of $\mathcal{D}$-module endofunctors of $\mathcal{M}$, whereby $\mathcal{M}$ becomes a $\mathcal{C}$-$\mathcal{D}$-bimodule category. We think of $M_\Sigma$ as a cobordism

$$M_\Sigma: \quad \emptyset \to \Sigma, \tag{50}$$

from the empty set to the gluing boundary $\Sigma \times \{1\}$. To the empty set $\emptyset$ the theory $\mathrm{TFT}_{\mathcal{D}}$ assigns the one-dimensional vector space $\mathbb{C}$. Applying $\mathrm{TFT}_{\mathcal{D}}$ to the cobordism (50) thus yields a linear map

$$\mathrm{TFT}_{\mathcal{D}}(M_\Sigma): \quad \mathbb{C} \to \mathcal{H}_\Sigma^0. \tag{51}$$

To emphasize the difference between the gluing and the physical boundaries, it is perhaps useful to consider what happens when we take both boundaries to be gluing boundaries. In this case, we have a cobordism $M_\Sigma'$ from one gluing boundary to the other; applying $\mathrm{TFT}_{\mathcal{D}}$ to this cobordism yields a linear map $\mathrm{TFT}_{\mathcal{D}}(M_\Sigma'): \mathcal{H}_\Sigma^0 \to \mathcal{H}_\Sigma^0$. In contrast to the map (51), which is described by the vector $\mathrm{TFT}_{\mathcal{D}}(M_\Sigma)(1) \in \mathcal{H}_\Sigma^0$, the map $\mathrm{TFT}_{\mathcal{D}}(M_\Sigma')$ is an operator which acts on such a vector, and the corresponding tensor network object would be a projected entangled

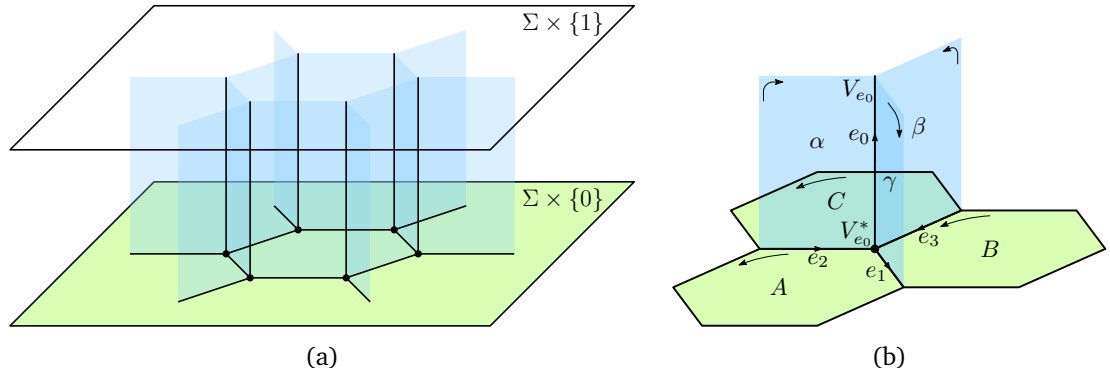

Figure 3: (a) A region of the three-manifold $M_\Sigma$, where the surface $\Sigma$ is endowed with a honeycomb-lattice cell decomposition. The physical boundary is depicted in green, the gluing boundary is white. (b) The assignment of state-sum variables to the two types of plaquettes with their orientation, as well as the vector spaces $V_{e_0}$ and $V_{e_0}^*$ associated to half-edges of $M_\Sigma$.

pair operator (PEPO). If instead we take both boundaries to be physical boundaries, then even though the resulting three-manifold $M_\Sigma''$ has a non-empty boundary, it has to be treated as a cobordism from the empty set to the empty set; applying $\mathrm{TFT}_{\mathcal{D}}$ to this cobordism yields a linear map $\mathrm{TFT}_{\mathcal{D}}\left(M_\Sigma''\right) : \mathbb{C} \to \mathbb{C}$ and thus a complex number. As a tensor network, the resulting object would represent the inner product between two PEPS; it would not have any physical degrees of freedom.

Once we are able to give an explicit description of $\mathrm{TFT}_{\mathcal{D}}$ on $M_\Sigma$, the map (51) provides us with a construction of a vector in the space $\mathcal{H}_\Sigma^0$. Such an explicit description is indeed provided by the Turaev-Viro construction. For performing this construction we need several additional ingredients. First, we fix a skeleton $P$ for the three-manifold $M_\Sigma$; the TFT will not depend on this skeleton, but to most directly recover the PEPS description we choose a skeleton consisting of prisms fitting with the cell decomposition $\Delta$ of $\Sigma$. The skeleton $P$ of $M_\Sigma$ is depicted in Figure 3; note that there are no vertices or edges on the gluing boundary. Secondly, we attach state-sum variables $\alpha, \beta, \gamma, \ldots \in I_{\mathcal{D}}$ to the oriented plaquettes of the skeleton that lie in the interior of $M_\Sigma$ (these are shaded blue in Figure 3b). There are also oriented plaquettes in the physical boundary (shaded green in Figure 3b); to these we assign state-sum variables $A, B, C, \ldots \in I_{\mathcal{M}}$, with $I_{\mathcal{M}}$ a set of representatives for the isomorphism classes of simple objects of $\mathcal{M}$. It is worth noting that there is an equivalent formulation of the Turaev-Viro construction in which the state-sum variables are attached to edges rather than plaquettes; this is related to the present construction by Poincaré duality. The formulation chosen here, which corresponds to the exposition in [36], connects more directly to the PEPS formalism.

We think of an edge of $P$ as consisting of two half-edges, each attached to one of the two ends of the edge. To every oriented half-edge in $P$ we associate a vector space given by a morphism space involving the state-sum variables (in $\mathcal{D}$ or in $\mathcal{M}$) that label the adjacent faces, with the distinction between domain and codomain determined by the orientations of the half-edge and of the faces. Specifically, if the edge is in the interior of $M_\Sigma$, such as $e_0$ in Figure 3b, then the three adjacent faces have state sum variables in $\mathcal{D}$, hence we have the three simple objects $\alpha, \beta \, \gamma$ in $\mathcal{D}$. Accordingly, the spaces assigned to the two half-edges are

$$\mathrm{Hom}_{\mathcal{D}}(\alpha \otimes \beta, \gamma) =: V_{e_0} \quad \text{and} \quad \mathrm{Hom}_{\mathcal{D}}(\gamma, \alpha \otimes \beta) \cong \mathrm{Hom}_{\mathcal{D}}(\alpha \otimes \beta, \gamma)^* = V_{e_0}^*, \tag{52}$$

respectively. For an edge on the physical boundary, such as the edges $e_1$, $e_2$ and $e_3$ in Figure 3b, two adjacent faces are on the physical boundary and thus have state sum variables $A, B$ in $\mathcal{M}$. The face pointing to the interior still has a label $\gamma$ in $\mathcal{D}$. Correspondingly, the spaces for

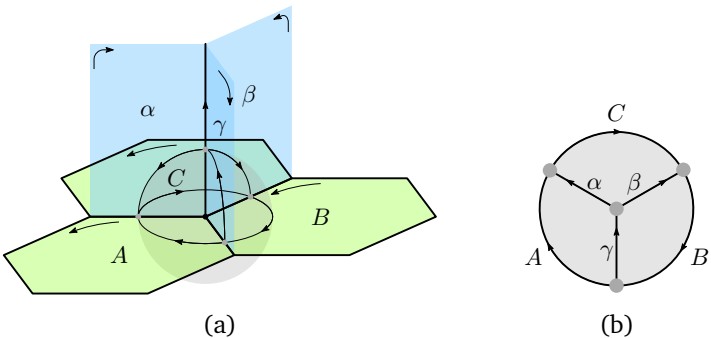

$$(a) \qquad\qquad\qquad (b)$$

Figure 4: Application of the evaluation map on a vertex of the physical boundary (a) leads to a tetrahedral diagram (b) that upon evaluation yields the PEPS tensor.

the half-edges are given by e.g.

$$\mathrm{Hom}_{\mathcal{M}}(A \triangleleft \gamma, B) =: V_{e_1} \quad \text{and} \quad \mathrm{Hom}_{\mathcal{M}}(B, A \triangleleft \gamma) \cong \mathrm{Hom}_{\mathcal{M}}(A \triangleleft \gamma, B)^* = V_{e_1}^*, \tag{53}$$

respectively. (Recall that the symbol $\triangleleft$ denotes the right action of $\mathcal{D}$ on $\mathcal{M}$.) Further, to every oriented edge $e \in P$ we associate the vector space $V_e \otimes V_e^*$ that is given by the tensor product of the two spaces attached to those of its two half-edges. Finally, to the three-manifold with skeleton $P$ we associate the big vector space

$$V_P := \bigotimes_{e \in P} V_e \otimes V_e^*. \tag{54}$$

Each of the tensor products $V_e \otimes V_e^*$ in $V_P$ is a tensor product of two vector spaces in a dual pairing and thus contains a canonical vector

$$v_e = \sum_i b_i \otimes b^i, \tag{55}$$

with $\{b_i\}$ any basis of $V_e$ and $\{b^i\}$ the dual basis of $V_e^*$. (In the tensor network language this canonical vector appears as the maximally entangled state used to contract two tensors, as in Eq. (66).) Using these canonical vectors, we find a canonical vector

$$v_P := \bigotimes_{e \in P} v_e \tag{56}$$

in $V_P$, which is independent of the specific choices of bases for the vector spaces $V_e$.

At every vertex $v$ of $P$ we have an evaluation map, which can be constructed as follows. For each vertex $v$ we draw a closed ball neighbourhood $B_v$ around $v$, as shown in Figure 4a. Then we construct a graph $\Gamma_v$ on the boundary of this ball by the following prescription: the edges of $\Gamma_v$ are defined as the intersection of the plaquettes of the skeleton $P$ with the boundary of $B_v$; they inherit their orientation and label from the orientation and state-sum variable of the plaquette. The intersections of the edges of $P$ with the boundary of $B_v$ give the vertices of $\Gamma_v$, to which we attach the vector space of the associated half-edge. This prescription yields a tetrahedral graph on a sphere, of the form depicted in Figure 4b. The resulting graph can be evaluated according to the rules of state-sum TFT as described e.g. in [36, Chapter 12.2], slightly extended such that the labels of $\Gamma_v$ are allowed to take values in the module category $\mathcal{M}$ in addition to the fusion category $\mathcal{D}$. Specifically, a tetrahedral graph evaluates to a $6j$ symbol, and in the case at hand this is indeed the $^3F$ symbol of Eq. (13c). The evaluation map for a vertex $v$ is thus a map from the tensor product of the vector spaces $V_e$ associated to the half-edges incident to $v$ to the complex numbers, e.g.

$$\mathrm{ev}_v: \quad V_{e_0}^* \otimes V_{e_1}^* \otimes V_{e_2} \otimes V_{e_3} \to \mathbb{C}, \tag{57}$$

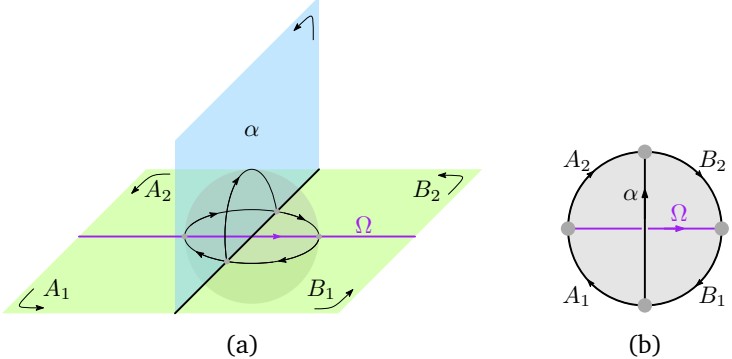

Figure 5: Application of the evaluation map to the intersection of the boundary Wilson line $\Omega$ with an edge on the physical boundary (a) leads to a tetrahedral diagram (b) that in special cases evaluates to the MPO symmetry or MPO intertwiner tensors. Note that the lines labeled by $\Omega$ and $\alpha$ do not cross, as is evident from (a).

for the vertex depicted in Figure 3b. Combining the evaluations for all vertices of $P$ gives a linear map

$$\mathrm{ev}_P = \bigotimes_{v \in P} \mathrm{ev}_v : \quad V_P \to \bigotimes_{\substack{e \text{ ending} \\ \text{on gluing} \\ \text{boundary}}} V_e = \mathcal{H}_\Sigma . \tag{58}$$

Applying this map to the vector (56) gives a state $\mathrm{ev}_P(v_P) \in \mathcal{H}_\Sigma$. Note that by this evaluation all vector spaces for half-edges that are connected to a vertex disappear. However, there are no vertices on the gluing boundary, hence the vector spaces assigned to the half-edges ending on gluing boundaries survive the evaluation. By inspection, one verifies that this state is indeed the PEPS associated with the data $\mathcal{D}$ and $\mathcal{M}$,

$$\mathrm{ev}_P(v_P) = \mathrm{PEPS}_{\mathcal{M},\mathcal{D}} . \tag{59}$$

By construction the state $\mathrm{ev}_P(v_P)$ lies in the space $\mathcal{H}_\Sigma^0$ that the TFT associates to the gluing boundary: it is the vector that the state-sum construction assigns to $M_\Sigma$, i.e. $\mathrm{TFT}_\mathcal{D}(M_\Sigma)(1)$, with $1 \in \mathbb{C} = \mathrm{TFT}_\mathcal{D}(\emptyset)$. Thus the equality (59) expresses in particular that the PEPS lies in the subspace $\mathcal{H}_\Sigma^0 \subset \mathcal{H}_\Sigma$ of ground states.

It is now fairly straightforward to also study the more general case that the physical boundary contains Wilson lines. In general, such a boundary Wilson line separates regions labeled by two different boundary conditions corresponding to different module categories $\mathcal{M}_1$ and $\mathcal{M}_2$. The boundary Wilson lines themselves are objects in the category $\mathrm{Fun}_\mathcal{D}(\mathcal{M}_1, \mathcal{M}_2)$ of $\mathcal{D}$-module functors. Taking $A_1, B_1, \ldots \in I_{\mathcal{M}_1}$ and $A_2, B_2, \ldots \in I_{\mathcal{M}_2}$ as labels for the plaquettes in the regions of the physical boundary that correspond to $\mathcal{M}_1$ and $\mathcal{M}_2$, respectively, we get the situation depicted in Figure 5a around the intersection of the boundary Wilson line $\Omega$ with an edge on the physical boundary. We can again construct an evaluation map, leading to the tetrahedral graph in Figure 5b. This graph can be evaluated directly in the following two special cases:

1. $\mathcal{M}_1 = \mathcal{M}_2 = \mathcal{M}$. In this case the boundary Wilson line is labeled by a simple object $\Omega \in \mathrm{Fun}_\mathcal{D}(\mathcal{M}, \mathcal{M}) = \mathcal{D}_\mathcal{M}^* = \mathcal{C}$ and the graph evaluates to an ${}^2F$ symbol, as in Equation (13b) for the MPO symmetry tensors.

2. $\mathcal{M}_1 = \mathcal{D}$ and $\mathcal{M}_2 = \mathcal{M}$. In this case the Wilson line is labeled by a simple object $\Omega \in \mathrm{Fun}_\mathcal{D}(\mathcal{D}, \mathcal{M}) = \mathcal{M}$ and the graph evaluates to an ${}^3F$ symbol as in Equation (24a) for the MPO intertwiner tensors.

To be equally explicit in the generic case, a further extension of recoupling theory is needed. Note that one can obtain objects of the functor category $\mathrm{Fun}_{\mathcal{D}}(\mathcal{M}_1, \mathcal{M}_2)$ through the composition

$$\mathcal{M}_2 \times \mathcal{M}_1^{\mathrm{op}} \simeq \mathrm{Fun}_{\mathcal{D}}(\mathcal{D}, \mathcal{M}_2) \times \mathrm{Fun}_{\mathcal{D}}(\mathcal{M}_1, \mathcal{D}) \to \mathrm{Fun}_{\mathcal{D}}(\mathcal{M}_1, \mathcal{M}_2) \tag{60}$$

of functors, but this composition is not essentially surjective, in general. Instead, one can make use of the fact that $\mathrm{Fun}_{\mathcal{D}}(\mathcal{M}_1, \mathcal{M}_2)$ is equivalent to the *relative Deligne product* $\mathcal{M}_2 \boxtimes_{\mathcal{D}} \mathcal{M}_1^{\mathrm{op}}$, as follows e.g. from the so-called module Eilenberg-Watts calculus [38]. However, to the best of our knowledge a (basis-dependent) description of this equivalence analogous to the use of $F$ symbols in the special cases above has not yet been worked out.

# 6 Conclusion and outlook

We have used the mathematical structure of a bimodule category to explore tensor network formulations of concepts in topologically ordered phases. We showed that the consistency conditions of having non-local MPO symmetries encoded by a fusion category $\mathcal{C}$ in a PEPS representation of a string-net based on a spherical fusion category $\mathcal{D}$ are equivalent to the pentagon equations of a $(\mathcal{C}, \mathcal{D})$-bimodule category $\mathcal{M}$, thereby classifying explicit representations of the PEPS and MPO tensors. These bimodule categories also allowed us to construct MPO intertwiners between different PEPS representations of the same string-net providing a generalisation of virtual gauge transformations between PEPS that describe the same state. An important conclusion to be drawn from these MPO intertwiners is that they relate equivalent PEPS tensors with possibly distinct virtual bond dimensions. This is in contrast to the situation for MPS, where the fundamental theorem dictates that two equivalent MPS have the same virtual bond dimension, and our results will contribute to the formulation of a general fundamental theorem of PEPS.

The PEPS representations and their MPO symmetries allow for a description of all ground states on a torus of some given string-net model through an explicit construction of the tube algebra, which we have only briefly touched upon in Section 2.2. This tube algebra gives a realisation of the monoidal center $Z(\mathcal{C})$; as such it also yields a description for the anyonic excitations of these models and allows us to characterise their topological features using the MPO symmetries [6]. The PEPS representations for boundaries and domain walls in string-net models introduced in this work can then be used to investigate mechanisms of anyon condensation and to characterise the properties of excitations living on these domain walls. While this has already been understood in the abstract diagrammatic language of category theory [9, 13], tensor networks allow for the numerical simulation of such systems. This is especially relevant in the context of error-correcting codes based on string-net models, where the explicit computation of properties such as error thresholds has proven to be difficult using traditional methods [39–41]. By extending PEPS representations for string-nets to include boundaries and domain walls, tensor network methods will provide a handle on these challenging problems.

The tensor network representations also provide a means to perturb these string-net ground states away from the RG fixed point, allowing us to study phase transitions between different topological orders [42–45]. These phase transitions are intimately linked with anyon condensation [46, 47] which, as already mentioned in the context of boundaries and domain walls, can be explicitly described using bimodule categories. It would be interesting to see what happens to the MPO symmetries and the associated tube algebras when we approach a phase transition, where certain PEPS representations and MPO symmetries will provide a more natural framework than others.

As mentioned before, a further generalisation of the various $F$-symbols is required to de-

scribe the more general case of MPO intertwiners and domain walls shown in Figures 1a and 2a. The relevant structure is that of a bicategory with three objects, which compared to the 2-object bicategories in this work comes with a total of 15 $F$-symbols satisfying 21 pentagon equations. These further generalisations of $F$-symbols and their pentagon equations should all have a natural interpretation in the tensor network language. Whether or not there is anything regarding these MPO intertwiners and domain walls that can be described by a 3-object bicategory but not already by combining 2-object bicategories is an open question that requires further research.

It should be appreciated that Turaev-Viro models allow more flexibility in the cell decompositions, at least as long as one works with a single spherical fusion category. It remains a challenge to understand the evaluation of more general graphs on a sphere and to obtain explicit expressions beyond $F$-symbols so as to extend the computational power of PEPS to more general cell decompositions.

Another natural extension of the bimodule categories used throughout this paper is to also include fermionic or superfusion categories [48] and associated superbimodule categories. The formalism of MPO symmetries for fermionic topological orders has been worked out in [49] for the case when the categories $\mathcal{C}$, $\mathcal{M}$ and $\mathcal{D}$ coincide. We expect that we can extend this to include superbimodule categories in the same way as we did for the bosonic case.

Finally, we point out that our results on tensor network descriptions of topologically ordered states in (2+1) dimensions have immediate relevance for 2-dimensional critical lattice systems as well. This is due to the fact that also in those systems, the presence of non-local symmetries described by MPOs is an essential feature [19]. Using a mapping known as the strange correlator [20, 50], any topologically ordered PEPS can be mapped directly to the partition function of a critical statistical mechanics model described by a conformal field theory (CFT) in the continuum limit. Many properties of the CFT such as topological defects, torus and cylinder partition functions and operator content can be readily understood as the image of corresponding concepts in topologically ordered PEPS under the strange correlator map [51]. The generalisations of such PEPS as described in this paper allow for an explicit realisation of the off-diagonal $D$ and $E$ type minimal models, where the intertwiners between different representations provide a lattice understanding of simple current extension or orbifolding [52].

## Coda

"Philosophically, the theory of tensor categories may perhaps be thought of as a theory of vector spaces or group representations without vectors." [Etingof, Gelaki, Nikshych and Ostrik, Tensor Categories, AMS, 2016] Pragmatically, the theory of matrix product operator symmetries may therefore be thought of as a theory of vector spaces and group representations without vectors, with vectors. [5]

## Acknowledgements

We would like to thank Dominic Williamson for commenting on the equivalence of the consistency equations of MPO symmetries with the pentagon equations of a bimodule category, as well as Robijn Vanhove, Norbert Schuch and Jacob Bridgeman for fruitful discussions.

---

[5]But indeed it is even more: it deeply utilises the fact that fusion categories have their own representation theory – module and bimodule categories, including its own Morita theory – which allows us to obtain a full overview over all MPO symmetries.

This work has received funding from the European Research Council (ERC) under the European Union's Horizon 2020 research and innovation programme (grant agreement No 715861 (ERQUAF) and 647905 (QUTE)). LL is supported by a PhD fellowship from the Research Foundation Flanders (FWO). JF is supported by VR under project no. 2017-03836. CS is partially supported by the Deutsche Forschungsgemeinschaft (DFG, German Research Foundation) under Germany's Excellence Strategy - EXC 2121 "Quantum Universe" - 390833306.

# A  Tensor networks

The purpose of this appendix is to provide a basic overview of pertinent tensor network concepts that are used in the main text. For more information, we refer to [4, 6, 22, 53–55].

## A.1  Matrix product states

Matrix product states furnish an efficient approximation to the ground states of local gapped Hamiltonians for one-dimensional lattices. A state of such a system having $N$ sites is an element of the vector space $\mathcal{H}_{\text{phys}}^{\otimes N}$, with $\mathcal{H}_{\text{phys}}$ the state space at each site, called the *physical Hilbert space*. A translation invariant matrix product state (MPS) for such a system with periodic boundary conditions is specified by a 3-index tensor $A$ according to

$$|\psi(A)\rangle := \sum_{j_1, j_2, \dots, j_N}^{d} \text{Tr}(A^{j_1} A^{j_2} \cdots A^{j_N}) |j_1\rangle |j_2\rangle \cdots |j_N\rangle, \tag{61}$$

where $d$ is the dimension of $\mathcal{H}_{\text{phys}} \cong \mathbb{C}^d$ and $\{|j\rangle\}$ is an orthonormal basis of $\mathcal{H}_{\text{phys}}$. One leg of the tensor $A$ takes values in $\mathcal{H}_{\text{phys}}$, while the other two take values in an auxiliary *virtual space* of dimension $D$. Thus for each $j = 1, 2, \dots, d$ the tensor $A$ defines a $D \times D$-matrix $A^j$: in (61) the trace over an $N$-fold product of these matrices is taken. Diagrammatically, (61) can be expressed as

$$|\psi(A)\rangle \;=\; \boxed{\begin{array}{ccccc} A & A & \cdots & A \\ | & | & & | \\ j_1 & j_2 & \cdots & j_N \end{array}}. \tag{62}$$

It is immediate from its definition that the MPS (61) is invariant under the substitution $A^j \mapsto X A^j X^{-1}$ with $X$ an arbitrary invertible $D \times D$-matrix. Such a substitution is called a *virtual gauge transformation*.

Upon forming linear combinations and matrix products the matrices $A^j$ for all values $j \in \{1, 2, \dots, d\}$ generate an algebra which is a subalgebra of the $D^2$-dimensional algebra of $D \times D$-matrices. An MPS is called *injective* if this subalgebra is the full $D^2$-dimensional matrix algebra. If an MPS is not injective, then there exist *invariant subspaces* of the full matrix algebra such that the corresponding orthogonal projectors $P_1, P_2, \dots$ satisfy $A^j P_r = P_r A^j P_r$ for all $j = 1, 2, \dots, d$ and all $r$, as well as a unitary virtual gauge transformation $U$ such that all $U P_r U^\dagger$ and all $U A^j U^\dagger$ are simultaneously block diagonal. For instance, in the case of 2 subspaces the gauge transformed MPS tensors take the form

$$\tilde{A}^j = U A^j U^\dagger = \begin{bmatrix} B^j & D^j \\ 0 & C^j \end{bmatrix}. \tag{63}$$

For periodic boundary conditions, the off-diagonal blocks $D^j$ do not contribute to the MPS $|\psi(\tilde{A})\rangle$. Without loss of generality we can therefore assume that $D^j = 0$. The matrices $\tilde{A}^j$ for the non-injective MPS then simply become direct sums $\tilde{A}^j = B^j \oplus C^j$ and the non-injective MPS

decomposes into a direct sum of two MPS, each of which in turn might be injective. If not, we can simply iterate the argument until the original non-injective MPS has turned into a direct sum of injective MPS.

The most important property of injective MPS is captured in the *fundamental theorem* of MPS [56]. This theorem states that two injective MPS characterized by tensors $A$ and $B$, respectively, yield the same state $|\psi(A)\rangle = |\psi(B)\rangle$ for arbitrary system size $N$ if and only if $A$ and $B$ are related by a virtual gauge transformation $X$, i.e. if and only if

$$\text{—} X \text{—} A \text{—} = \text{—} B \text{—} X \text{—} \tag{64}$$

This result is of central importance in the classification of MPS, e.g. it is the basis of the classification of SPT phases in 1-dimension using group cohomology.

## A.2 Projected Entangled Pair States

A more physical interpretation of MPS is provided by the following different yet equivalent way of constructing MPS, which also carries over to higher-dimensional systems. We consider again a translation invariant system with periodic boundary conditions. At each site with physical $d$-dimensional degree of freedom $j$, we place two $D$-dimensional virtual degrees of freedom with orthonormal basis $\{|i\rangle\}$, yielding a $D^2$-dimensional Hilbert space. This is indicated in the following picture:

$$
\cdots \quad
\begin{matrix} D & D \\ \bullet & \bullet \\ & j_1 \end{matrix} \quad
\begin{matrix} D & D \\ \bullet & \bullet \\ & j_2 \end{matrix} \quad
\begin{matrix} D & D \\ \bullet & \bullet \\ & j_3 \end{matrix} \quad
\begin{matrix} D & D \\ \bullet & \bullet \\ & j_4 \end{matrix} \quad
\cdots
\tag{65}
$$

We now maximally entangle all the pairs of qudits on neighbouring sites by projecting onto the maximally entangled state

$$|\alpha\rangle = \sum_{i=1}^{D} |i\rangle |i\rangle . \tag{66}$$

We depict this prescription as

$$
\begin{matrix}
|\alpha\rangle\langle\alpha| & |\alpha\rangle\langle\alpha| & |\alpha\rangle\langle\alpha| & |\alpha\rangle\langle\alpha| & |\alpha\rangle\langle\alpha| \\
\cdots \bullet & \bullet \bullet & \bullet \bullet & \bullet \bullet & \bullet \cdots \\
j_1 & j_2 & j_3 & j_4 &
\end{matrix}
\tag{67}
$$

where the dotted lines indicate periodic boundary conditions. Finally, given a PEPS tensor $A$, at each site we act on the pair of qudits associated to it with the corresponding linear map $f_A \colon \mathbb{C}^D \otimes \mathbb{C}^D \to \mathbb{C}^d$, whereby the virtual degrees of freedom at the site are mapped to the the $d$-dimensional space of physical degrees of freedom:

$$
\begin{matrix}
\cdots \bullet \bullet & \bullet \bullet & \bullet \bullet & \bullet \bullet \cdots \\
j_1 & j_2 & j_3 & j_4
\end{matrix}
\tag{68}
$$

The so obtained state in $\mathcal{H}_{\text{phys}}^{\otimes N}$ is called a *projected entangled pair state*, or PEPS for short.

An attractive feature of this construction is that it has a straightforward generalisation to higher dimensions. Actually, the term PEPS is usually reserved for a PEPS in 2 dimensions; in Figure 6a we display such a PEPS. Similar to the MPS case, one can alternatively just define a PEPS as a tensor with one physical and $\ell$ virtual legs, as depicted for $\ell = 4$ in Figure 6a. The two constructions are equivalent.

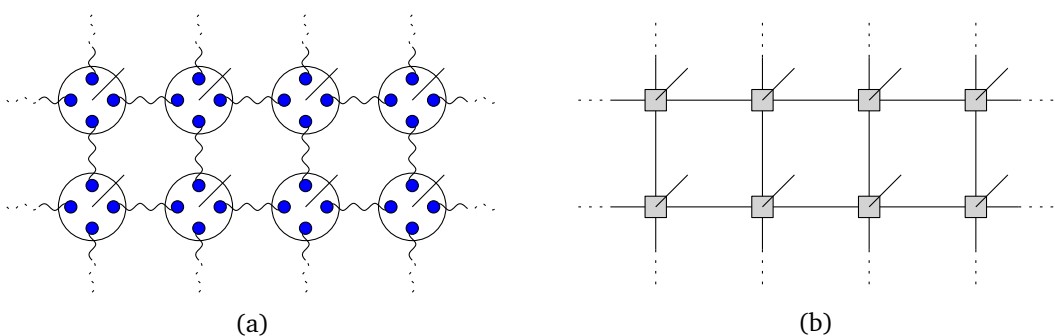

Figure 6: (a) A $2 \times 4$ patch of a 2-dimensional PEPS lattice, with the physical index drawn as sticking out up and to the right. (b) The same patch with more general tensors.

## A.3  Matrix product operators

Similarly as for MPS one can define translation invariant *matrix product operators* (MPO) with periodic boundary conditions, specified by a 4-index tensor $B$ with two $d_i$-dimensional internal and two $d_e$-dimensional external legs according to

$$
\hat{O}(B) = \sum_{\{i\},\{i'\}=1}^{d_e} \mathrm{Tr}\left(B^{i_1 i'_1} \cdots B^{i_n i'_n}\right) |i_1 \cdots i_n\rangle \langle i'_1 \cdots i'_n|
$$

$$
= \quad \boxed{\begin{array}{ccccc} i_1 & i_2 & \dots & i_n \\ B & B & \dots & B \\ i'_1 & i'_2 & \dots & i'_n \end{array}} . \tag{69}
$$

These MPOs represent a linear map $\mathbb{C}^{\otimes d_e^n} \to \mathbb{C}^{\otimes d_e^n}$. Taking $d_e = d$, these maps can be interpreted as operators acting on a suitable MPS with physical dimension $d$ or as a density matrix. In the present paper we use them instead exclusively as operators acting on the *virtual* degrees of freedom of a PEPS according to

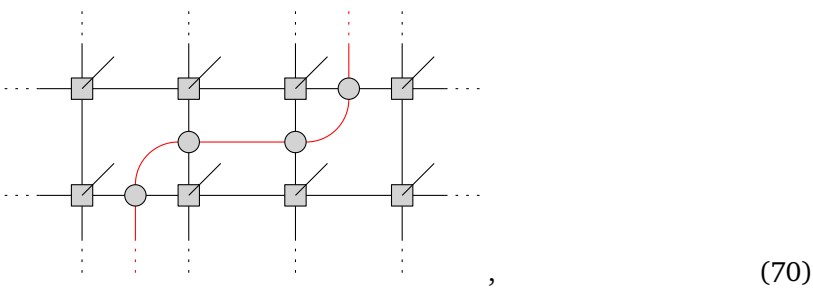

$$
, \tag{70}
$$

i.e. the external legs of the MPO are contracted with the virtual legs of the PEPS, so that in particular we have $d_e = D$. Here the internal legs of the MPO are drawn in red in order to distinguish them from the virtual legs of the PEPS.

Analogously to an MPS, an MPO is called injective if the $d_i \times d_i$-matrices $B^{ii'}$ with $i, i' \in \{1, 2, \dots, d_e\}$ generate the full $d_i^2$-dimensional matrix algebra. If the MPO is not injective, there exist invariant subspaces $P_a$ and a unitary virtual gauge transformation $U$ such that all $U P_a U^\dagger$ and all $U B^{ii'} U^\dagger$ are simultaneously block diagonal and the MPO matrices become a direct sum of injective MPO matrices:

$$
U B^{ii'} U^\dagger = \bigoplus_a B_a^{ii'} , \tag{71}
$$

where $B_a^{ii'}$ denote the different injective blocks. Moreover, owing to the periodic boundary conditions we can again assume any off-diagonal blocks to be zero. In the main text we denote the different injective MPOs as $\hat{O}_a := \hat{O}(B_a)$, and we keep track of the MPO injective block label $a$ as the label of the red line.

## A.4 Diagrammatic notation

In the situation studied in the main text we deal with a honeycomb lattice, so that the number of virtual legs of the PEPS tensor is $\ell = 3$. Also, in order to render the diagrammatic description of tensors unambiguous, the legs of all tensors must be oriented. Further, for visual clarity, we have made the following diagrammatic simplifications in the main text:

$$
\equiv \qquad \text{and} \qquad \equiv \qquad . \tag{72}
$$

In the explicit expressions for the fusion, MPO and PEPS tensors we are using a generalisation of the *triple line notation* that was introduced in [2, 3]. To relate this notation to more conventional tensor network notation, it suffices to unite the triple lines and their multiplicity together into a single index. It is also worth noting that we only list the non-zero components of the tensors. Hereby we avoid the proliferation of factors of Kronecker deltas for lines shared between two indices that results from the conventions in e.g. [2–4]. Finally, we do not orient the outer lines of the triple line since these are being summed over when contracting tensors, and the fact that there exists a consistent orientation of these lines justifies its omission. When applied to the fusion tensor $X_m$, this prescription amounts to the identification

$$
(cCA, n) \equiv c \rightarrow m \stackrel{a}{\underset{b}{\longrightarrow}}, \tag{73}
$$

where on the right hand side we only keep the MPO injective block labels $a, b, c$ with degeneracy label $m$, and where

$$
|cCA, n\rangle \in \mathrm{Hom}_{\mathcal{M}}(c \triangleright C, A),
$$
$$
|aBA, j\rangle \in \mathrm{Hom}_{\mathcal{M}}(a \triangleright B, A),
$$
$$
|bCB, k\rangle \in \mathrm{Hom}_{\mathcal{M}}(b \triangleright C, B). \tag{74}
$$

For the MPO tensor the identification is given by

$$
(aCA, j) \equiv \stackrel{\alpha}{\underset{a}{\uparrow}}, \tag{75}
$$

where on the right hand side we only keep the MPO injective block label $a$ and the PEPS injective block label $\alpha$, and

$$|aCA,j\rangle \in \mathrm{Hom}_{\mathcal{M}}(a \triangleright C, A),$$

$$|aDB,k\rangle \in \mathrm{Hom}_{\mathcal{M}}(a \triangleright D, B),$$

$$|C\alpha D,n\rangle \in \mathrm{Hom}_{\mathcal{M}}(C \triangleleft \alpha, D),$$

$$|A\alpha B,m\rangle \in \mathrm{Hom}_{\mathcal{M}}(A \triangleleft \alpha, B). \tag{76}$$

Finally, for the PEPS tensor we have

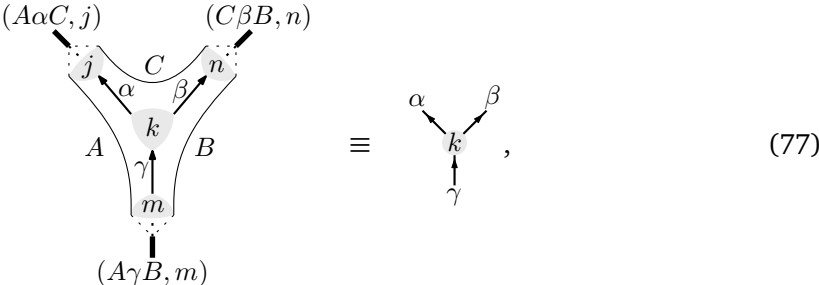

$$\tag{77}$$

where on the right hand side we only keep the PEPS injective block labels $\alpha, \beta, \gamma$ with degeneracy label $k$ and

$$|A\alpha C,j\rangle \in \mathrm{Hom}_{\mathcal{M}}(A \triangleleft \alpha, C),$$

$$|C\beta B,n\rangle \in \mathrm{Hom}_{\mathcal{M}}(C \triangleleft \beta, B),$$

$$|A\gamma B,m\rangle \in \mathrm{Hom}_{\mathcal{M}}(A \triangleleft \gamma, B). \tag{78}$$

Further, the physical leg of the PEPS – which is sticking out of the page – is labeled by $(\alpha\beta\gamma, k)$, with

$$|\alpha\beta\gamma,k\rangle \in \mathrm{Hom}_{\mathcal{D}}(\alpha \otimes \beta, \gamma). \tag{79}$$

# B  Categories

## B.1  Fusion categories

A *monoidal category* $(\mathcal{C}, \otimes, {}^0F, \mathbf{1})$ is a category $\mathcal{C}$ with a functor $\otimes : \mathcal{C} \times \mathcal{C} \to \mathcal{C}$ and a natural isomorphism ${}^0F : (a \otimes b) \otimes c \xrightarrow{\cong} a \otimes (b \otimes c)$ for $a, b, c \in \mathcal{C}$, called the *associator* or associativity constraint, and with a distinguished *unit object* $\mathbf{1} \in \mathcal{C}$ and natural isomorphisms $a \otimes \mathbf{1} \xrightarrow{\cong} a$ and $\mathbf{1} \otimes a \xrightarrow{\cong} a$, called the right and left unit constraints. The associator ${}^0F$ is required to satisfy the *pentagon relation*, which we will display explicitly in Section B.4. In addition there are two triangle relations involving the associator and the right and left unit constraint, respectively. Without loss of generality however, we take the unit object to be strict, meaning that the unit constraints are identities, so that $a \otimes \mathbf{1} = a = \mathbf{1} \otimes a$ on the nose and the triangle relations are trivial.

A *rigid monoidal category* is a monoidal category for which every object $a$ has a left dual ${}^{\vee}a$ and a right dual $a^{\vee}$ and there are left evaluation and coevaluation morphisms $\widetilde{\mathrm{ev}}_a : a \otimes {}^{\vee}a \to \mathbf{1}$ and $\widetilde{\mathrm{coev}}_a : \mathbf{1} \to {}^{\vee}a \otimes a$ and right evaluation and coevaluation morphisms $\mathrm{ev}_a : a^{\vee} \otimes a \to \mathbf{1}$ and $\mathrm{coev}_a : \mathbf{1} \to a \otimes a^{\vee}$, required to satisfy the so-called snake identities. We represent the left and right evaluation and coevaluation graphically as

$$\widetilde{\mathrm{ev}}_a = \overset{a \quad {}^{\vee}a}{\underset{}{\cup}}, \qquad \widetilde{\mathrm{coev}}_a = \underset{{}^{\vee}a \quad a}{\overset{\frown}{}}, \qquad \mathrm{ev}_a = \overset{a^{\vee} \quad a}{\underset{}{\cup}}, \qquad \mathrm{coev}_a = \underset{a \quad a^{\vee}}{\overset{\frown}{}}. \tag{80}$$

Then the snake identities look as follows:

$$a \overset{\frown}{\underset{a}{}} \underset{\frown}{} a = \left|\begin{array}{c} a \\ a \end{array}\right. = a \overset{\frown}{} \underset{\underset{a}{\frown}}{} , \qquad a^\vee \overset{\frown}{\underset{\frown}{}} a^\vee = \left|\begin{array}{c} a^\vee \\ a^\vee \end{array}\right. , \qquad {}^\vee a \overset{\frown}{} \underset{\frown}{} {}^\vee a = \left|\begin{array}{c} {}^\vee a \\ {}^\vee a \end{array}\right. . \tag{81}$$

A *fusion category* is a semi-simple rigid monoidal category which is in addition *linear* and satisfies certain finiteness conditions (for a precise definition see e.g. Chapter 4.1 of [57]). In particular, the morphism sets $\mathrm{Hom}_\mathcal{C}$ are finite-dimensional vector spaces over some field, which for our purposes is the complex numbers $\mathbb{C}$, and the number of isomorphism classes of simple objects is finite. We select a set $I_\mathcal{C}$ of representatives for these classes that contains $\mathbf{1}$. In terms of the simple objects in this set we have

$$a \otimes b \cong \bigoplus_{c \in I_\mathcal{C}} N_{ab}^c \, c \,. \tag{82}$$

The multiplicities $N_{ab}^c = \dim_\mathbb{C}(\mathrm{Hom}_\mathcal{C}(a \otimes b, c))$ are called the fusion rules of $\mathcal{C}$; they satisfy

$$\sum_{e \in I_\mathcal{C}} N_{ab}^e N_{ec}^d = \sum_{f \in I_\mathcal{C}} N_{af}^d N_{bc}^f \tag{83}$$

and $N_{a\mathbf{1}}^b = \delta_a^b = N_{\mathbf{1}a}^b$. The associator ${}^0F$ and its inverse, which we denote as ${}_0F$, can be expressed in terms of the simple objects in $I_\mathcal{C}$ as follows (using the diagrammatic language for morphisms, to be read from top to bottom):

$$\tag{84}$$

Here $m, n$ and $j, k$ are multiplicity labels, for instance $j$ labels a basis vector in $\mathrm{Hom}_\mathcal{C}(a \otimes b, e)$. By definition, ${}^0F$ and ${}_0F$ are indeed inverses, in the sense that

$$\sum_{f, mn} \left({}^0F_d^{abc}\right)_{e,jk}^{f,mn} \left({}_0F_d^{abc}\right)_{e',j'k'}^{f,mn} = \delta_{ee'} \delta_{jj'} \delta_{kk'} \,. \tag{85}$$

Furthermore, the basis vectors in the one-dimensional vector spaces $\mathrm{Hom}_\mathcal{C}(a \otimes \mathbf{1}, a)$ and $\mathrm{Hom}_\mathcal{C}(\mathbf{1} \otimes a, a)$ can be chosen such that

$$\left({}^0F_d^{\mathbf{1}bc}\right)_{b,1k}^{d,m1} = \left({}^0F_d^{a\mathbf{1}c}\right)_{a,1k}^{c,1m} = \left({}^0F_d^{ab\mathbf{1}}\right)_{d,k1}^{b,1m} = \delta_k^m \,. \tag{86}$$

In our context, we have to deal with two different fusion categories. We denote the second one by $\mathcal{D}$, the elements of the finite set $I_\mathcal{D}$ of simple objects by $\alpha, \beta, \dots$, and the fusion rules of $\mathcal{D}$ by

$$\alpha \otimes \beta \cong \bigoplus_{\gamma \in I_\mathcal{D}} N_{\alpha\beta}^\gamma \, \gamma, \tag{87}$$

with $N_{\alpha\beta}^\gamma = \dim_\mathbb{C}(\mathrm{Hom}_\mathcal{D}(\alpha \otimes \beta, \gamma))$, satisfying $\sum_\mu N_{\alpha\beta}^\mu N_{\mu\gamma}^\delta = \sum_\nu N_{\alpha\nu}^\delta N_{\beta\gamma}^\nu$. We denote the associator of $\mathcal{D}$ by ${}^4F$ and its inverse by ${}_4F$; they satisfy

$$\tag{88}$$

We denote the unit object of $\mathcal{D}$ again by $\mathbf{1}$; again we can make basis choices such that

$$\left({}^4F_\delta^{\mathbf{1}\beta\gamma}\right)_{\beta,1k}^{\delta,m1} = \left({}^4F_\delta^{\alpha\mathbf{1}\gamma}\right)_{\alpha,1k}^{\gamma,1m} = \left({}^4F_\delta^{\alpha\beta\mathbf{1}}\right)_{\delta,k1}^{\beta,1m} = \delta_k^m. \tag{89}$$

## B.2 Module categories

A left *module category* $(\mathcal{M}, \triangleright, {}^1F)$ over a fusion category $(\mathcal{C}, \otimes, {}^0F)$ is a (linear, semisimple) category $\mathcal{M}$ with a functor $\triangleright : \mathcal{C} \times \mathcal{M} \to \mathcal{M}$ (called the *action* of $\mathcal{C}$ on $\mathcal{M}$) and a natural isomorphism ${}^1F : (a \otimes b) \triangleright A \xrightarrow{\cong} a \triangleright (b \triangleright A)$ with $a, b \in \mathcal{C}$ and $A \in \mathcal{M}$ that satisfies a mixed pentagon relation. We take the action of the (strict) unit object $\mathbf{1} \in \mathcal{C}$ to be strict, i.e. $\mathbf{1} \triangleright A = A$.

We select a set $I_{\mathcal{M}}$ of representatives for the isomorphism classes of simple objects of $\mathcal{M}$ and write

$$a \triangleright A \cong \bigoplus_{B \in I_{\mathcal{M}}} N_{aA}^B B, \tag{90}$$

with $N_{aA}^B = \dim_{\mathbb{C}}(\mathrm{Hom}_{\mathcal{M}}(a \triangleright A, B))$, satisfying

$$\sum_{c \in I_{\mathcal{C}}} N_{ab}^c N_{cA}^B = \sum_{C \in I_{\mathcal{M}}} N_{aC}^B N_{bA}^C. \tag{91}$$

The isomorphism ${}^1F$ and its inverse ${}_1F$ can be expressed as follows:

$$\tag{92}$$

We can choose bases in the one-dimensional morphisms spaces involving the unit object in such a way that

$$\left({}^1F_B^{\mathbf{1}bA}\right)_{b,1k}^{B,m1} = \left({}^1F_B^{a\mathbf{1}A}\right)_{a,1k}^{A,1m} = \delta_k^m. \tag{93}$$

Analogously, a *right* module category $(\mathcal{M}, \triangleleft, {}^3F)$ over the fusion category $\mathcal{D}$ is a category $\mathcal{M}$ with a right action functor $\triangleleft : \mathcal{M} \times \mathcal{D} \to \mathcal{M}$ and a natural isomorphism ${}^3F : (A \triangleleft \alpha) \triangleleft \beta \xrightarrow{\cong} A \triangleleft (\alpha \otimes \beta)$ with $A \in \mathcal{M}$ and $\alpha, \beta \in \mathcal{D}$. In terms of the sets $I_{\mathcal{D}}$ and $I_{\mathcal{M}}$ simple objects, the right action $\triangleleft$ is expressed as

$$A \triangleleft \alpha \cong \bigoplus_{B \in I_{\mathcal{M}}} N_{A\alpha}^B B, \tag{94}$$

with $N_{A\alpha}^B = \dim_{\mathbb{C}}(\mathrm{Hom}_{\mathcal{M}}(A \triangleleft \alpha, B))$ satisfying

$$\sum_{C \in I_{\mathcal{M}}} N_{A\alpha}^C N_{C\beta}^B = \sum_{\gamma \in I_{\mathcal{D}}} N_{A\gamma}^B N_{\alpha\beta}^\gamma, \tag{95}$$

while the isomorphism ${}^3F$ and its inverse ${}_3F$ are described as

$$\tag{96}$$

Again we can choose bases such that

$$\left({}^3F_B^{A\mathbf{1}\beta}\right)_{A,1k}^{\beta,1m} = \left({}^3F_B^{A\alpha\mathbf{1}}\right)_{B,k1}^{\alpha,1m} = \delta_k^m. \tag{97}$$

### B.3 Bimodule categories

A $(\mathcal{C}, \mathcal{D})$-bimodule category $(\mathcal{M}, \triangleright, \triangleleft, {}^1F, {}^3F, {}^2F)$ over a pair of fusion categories $\mathcal{C}$ and $\mathcal{D}$ is a category $\mathcal{M}$ with additional structure such that $(\mathcal{M}, \triangleright, {}^1F)$ is a left $\mathcal{C}$-module category and $(\mathcal{M}, \triangleleft, {}^3F)$ is a right $\mathcal{D}$-module category and such that there is a natural isomorphism ${}^2F : (a \triangleright A) \triangleleft \alpha \xrightarrow{\cong} a \triangleright (A \triangleleft \alpha)$ for $a \in \mathcal{C}$, $A \in \mathcal{M}$ and $\alpha \in \mathcal{D}$. In terms of simple objects, this imposes the compatibility condition

$$\sum_C N_{aA}^C N_{C\alpha}^B = \sum_D N_{aD}^B N_{A\alpha}^D \tag{98}$$

on the left and right action functors. The isomorphism ${}^2F$ and its inverse ${}_2F$ give

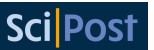

$$\left({}^2F_B^{1A\alpha}\right)_{A,1k}^{\alpha,m1} = \left({}^2F_B^{aA1}\right)_{B,k1}^{A,1m} = \delta_k^m. \tag{100}$$

### B.4 Pentagon equations

The natural isomorphisms ${}^0F$, ${}^1F$, ${}^2F$, ${}^3F$ and ${}^4F$ for a pair of fusion categories $\mathcal{C}$ and $\mathcal{D}$ and a $(\mathcal{C}, \mathcal{D})$-bimodule category $\mathcal{M}$ as described above satisfy coupled consistency conditions known as pentagon equations. They are a core ingredient of this paper, and therefore we present all of them in this separate section. One of these equations is the pentagon equation for the fusion category $\mathcal{C}$; it expresses the equality of the two ways in which an isomorphism

$$((a \otimes b) \otimes c) \otimes d \xrightarrow{\cong} a \otimes (b \otimes (c \otimes d))$$

can be constructed from the associator ${}^0F$ of the category. It reads (see formula (6) in the main text)

$$\sum_o \left({}^0F_e^{fcd}\right)_{g,lm}^{h,no} \left({}^0F_e^{abh}\right)_{f,ko}^{i,pq} = \sum_{j,rst} \left({}^0F_g^{abc}\right)_{f,kl}^{j,rs} \left({}^0F_e^{ajd}\right)_{g,sm}^{i,tq} \left({}^0F_i^{bcd}\right)_{j,rt}^{h,np}, \tag{$P_0$}$$

an equality to which we will refer as $P_0$. Similarly, equating the two ways in which an isomorphism

$$((a \otimes b) \otimes c) \triangleright A \xrightarrow{\cong} a \triangleright (b \triangleright (c \triangleright A))$$

can be constructed using ${}^0F$ and ${}^1F$ gives

$$\sum_o \left({}^1F_B^{fcA}\right)_{g,lm}^{C,no} \left({}^1F_B^{abC}\right)_{f,ko}^{D,pq} = \sum_{j,rst} \left({}^0F_g^{abc}\right)_{f,kl}^{j,rs} \left({}^1F_B^{ajA}\right)_{g,sm}^{D,tq} \left({}^1F_D^{bcA}\right)_{j,rt}^{C,np}, \tag{$P_1$}$$

which we call $P_1$, the analogous procedure for

$$((a \otimes b) \triangleright A) \triangleleft \alpha \xrightarrow{\cong} a \triangleright (b \triangleright (A \triangleleft \alpha))$$

gives

$$\sum_o \left({}^2F_B^{fA\alpha}\right)_{C,lm}^{D,no} \left({}^1F_B^{abD}\right)_{f,ko}^{E,pq} = \sum_{F,rst} \left({}^1F_C^{abA}\right)_{f,kl}^{F,rs} \left({}^2F_B^{aF\alpha}\right)_{C,sm}^{E,tq} \left({}^2F_E^{bA\alpha}\right)_{F,rt}^{D,np}, \tag{$P_2$}$$

which we call $P_2$,

$$((a \rhd A) \lhd \alpha) \lhd \beta \xrightarrow{\cong} a \rhd (A \lhd (\alpha \otimes \beta))$$

gives

$$\sum_o \left({}_3F_B^{C\alpha\beta}\right)_{D,lm}^{\gamma,no} \left({}_2F_B^{aA\gamma}\right)_{C,ko}^{E,pq} = \sum_{F,rst} \left({}_2F_D^{aAa}\right)_{C,kl}^{F,rs} \left({}_2F_B^{aF\beta}\right)_{D,sm}^{E,tq} \left({}_3F_E^{A\alpha\beta}\right)_{F,rt}^{\gamma,np}, \qquad (P_3)$$

which we call $P_3$, and

$$((A \lhd \alpha) \lhd \beta) \lhd \gamma \xrightarrow{\cong} A \lhd (\alpha \otimes (\beta \otimes \gamma))$$

gives

$$\sum_o \left({}_3F_B^{C\beta\gamma}\right)_{D,lm}^{\mu,no} \left({}_3F_B^{A\alpha\mu}\right)_{C,ko}^{\nu,pq} = \sum_{\delta,rst} \left({}_3F_D^{A\alpha\beta}\right)_{C,kl}^{\delta,rs} \left({}_3F_B^{A\delta\gamma}\right)_{D,sm}^{\nu,tq} \left({}_4F_\nu^{\alpha\beta\gamma}\right)_{\delta,rt}^{\mu,np}, \qquad (P_4)$$

which we call $P_4$. Finally

$$((\alpha \otimes \beta) \otimes \gamma) \otimes \delta \xrightarrow{\cong} \alpha \otimes (\beta \otimes (\gamma \otimes \delta))$$

gives

$$\sum_o \left({}_4F_\rho^{\eta\gamma\delta}\right)_{\lambda,lm}^{\mu,no} \left({}_4F_\rho^{\alpha\beta\mu}\right)_{\eta,ko}^{\nu,pq} = \sum_{\kappa,rst} \left({}_4F_\lambda^{\alpha\beta\gamma}\right)_{\eta,kl}^{\kappa,rs} \left({}_4F_\rho^{\alpha\kappa\delta}\right)_{\lambda,sm}^{\nu,tq} \left({}_4F_\nu^{\beta\gamma\delta}\right)_{\kappa,rt}^{\mu,np}, \qquad (P_5)$$

which is the pentagon identity for the fusion category $\mathcal{D}$ and which we call $P_5$. The inverses ${}_0F$, ${}_1F$, ${}_2F$, ${}_3F$ and ${}_4F$ satisfy a very similar system of coupled pentagon equations. They can be derived in parallel with those above, and we refrain from presenting them here.

## C Pivotal and spherical structure

### C.1 Pivotal, spherical and unitary fusion categories

In a rigid monoidal category $\mathcal{C}$ the evaluation and coevaluation morphisms can be concatenated as done in the snake identities (81). But they cannot, in general, be composed so as to obtain endomorphisms of the tensor unit, i.e. numbers. Rather, to be able to do so, we must have a way to consistently identify the left and right dual of an object or, equivalently, an object and its double dual. To be precise, what is needed is a monoidal natural isomorphism between the identity functor and the double dual functor of $\mathcal{C}$. Such a natural isomorphism is called a *pivotal structure*, and a rigid monoidal category with a pivotal structure is called a pivotal category. For a pivotal fusion category we can identify left and right duals, so that for any $a \in \mathcal{C}$ we only need to deal with a single dual object, which we denote by $\bar{a}$; an object $a$ is called *self-dual* if $\bar{a} \cong a$. Moreover, every pivotal fusion category is equivalent, as a pivotal category, to one in which the pivotal structure is trivial, so that in particular $\bar{\bar{a}} = a$; we will tacitly assume that we work with this equivalent fusion category.

In a pivotal category we can define two traces of an endomorphism $f \in \mathrm{Hom}_{\mathcal{C}}(a, a)$ by

$$\boxed{f} \quad \text{and} \quad \boxed{f} \qquad (101)$$

respectively, and thus in particular two dimensions of an object $a$ as traces of the identity morphism $\mathrm{id}_a$. If the two traces coincide for every endomorphim $f$, then the pivotal category is called *spherical*. For a spherical fusion category $\mathcal{C}$ we denote the (unique) dimension of $a \in \mathcal{C}$ by $d_a$.

A $\mathbb{C}$-linear fusion category is called *unitary* if it comes with an involutive antilinear contravariant endofunctor $\dagger$ that is the identity on objects, is compatible with tensor products, and such that the morphism spaces are Hilbert spaces, and if the associativity constraint is unitary.

In a unitary fusion category one can choose the bases of the morphism spaces used in the definition (84) of the $^0F$-symbols in such a way that the $^0F$-symbols form unitary matrices. (Conversely, if this is possible, then the fusion category is unitary.) A unitary fusion category has a canonical pivotal structure which is even spherical. With respect to this pivotal structure the dimension $d_a$ of every object is positive. We will often take square roots of these numbers, and one must keep track of the choice of square root. Taking the fusion category to be unitary makes this rather straightforward as we can always choose the positive root, as done in the main text. We note that unitarity is not a necessary requirement and that the generalisation of the MPO formalism to nonunitary fusion categories is straightforward [58].

We take the normalisation of the basis vectors of $\mathrm{Hom}_{\mathcal{C}}(a \otimes b, c)$ and $\mathrm{Hom}_{\mathcal{C}}(c, a \otimes b)$ to be such that

$$a \left\langle\!\!\!\!\!\underset{j}{\overset{k}{\diamondsuit}}\!\!\!\!\!\right\rangle b = \delta_{j,k}\delta_{c,c'}\sqrt{\frac{d_a d_b}{d_c}} \; \Bigg|_c^c \; , \qquad \sum_{c,j}\sqrt{\frac{d_c}{d_a d_b}}\;\; = \;\; \Bigg|_a^a \, \Bigg|_b^b \; . \tag{102}$$

This choice facilitates some of the explicit formulas that we will give below; it is worth noting that different conventions are in use as well.

In a $\mathbb{C}$-linear spherical fusion category one has

$$\left(^0F_a^{a\bar{a}a}\right)^{1,11}_{1,11} = \frac{\varkappa_a}{d_a}, \tag{103}$$

with $\varkappa_a = \varkappa_{\bar{a}}^*$ a phase. One can make a consistent gauge choice such that $\varkappa_a = 1$ for every non-selfdual simple object $a$. In contrast, for simple objects $a$ that are self-dual, the number $\left(^0F_a^{a\bar{a}a}\right)^{1,11}_{1,11}$ is a gauge invariant quantity, and hence so is $\varkappa_a$. One can show that in this case $\varkappa_a \in \{1, -1\}$; this number is known as the *Frobenius-Schur indicator* of $a$. The number $\varkappa_a$ can be expressed in terms of a suitable endomorphism of $a$, which diagrammatically looks as follows:

$$a \left\langle \overset{a}{\underset{1}{\diamondsuit}} \bar{a} \right\rangle a = \left(^0F_a^{a\bar{a}a}\right)^{1,11}_{1,11} a \left\langle \overset{a}{\underset{1}{\diamondsuit}} \bar{a} \right\rangle a = d_a\left(^0F_a^{a\bar{a}a}\right)^{1,11}_{1,11} \Bigg|_a \Bigg|_a = \varkappa_a \Bigg|_a \Bigg| . \tag{104}$$

Here we do not write any multiplicity labels, as the morphism spaces involved are all one-dimensional. In the following we will make a particular choice for the left and right evaluation and coevaluation morphisms:

$$a \,\overset{\frown}{\underset{}{\smile}}\, \bar{a} := \overset{a \quad \bar{a}}{\underset{1}{\vee}} \, , \qquad \bar{a} \,\overset{\frown}{\underset{}{\frown}}\, a := \varkappa_a^* \overset{1}{\underset{\bar{a} \quad a}{\wedge}} \, , \tag{105}$$

$$\bar{a} \,\overset{\frown}{\underset{}{\smile}}\, a := \varkappa_a \overset{\bar{a} \quad a}{\underset{1}{\vee}} \, , \qquad a \,\overset{\frown}{\underset{}{\frown}}\, \bar{a} := \overset{1}{\underset{a \quad \bar{a}}{\wedge}} \, . \tag{106}$$

Using Eq. (104), these can easily be seen to satisfy the snake identities (81).

## C.2 MPO symmetries

In the explicit identifications of the various tensors in the main text with $F$-symbols there are various factors of (powers of) quantum dimensions. Of these, the ones that are appear with a power $1/4$ are simply normalization choices to be consistent with Eq. (102). The other factors appear as a square root, and they are chosen such that the properties pertaining to sphericity of the relevant fusion categories are satisfied at the level of the tensors as well. To illustrate this for the MPO symmetries, we start by noticing that due to the gauge choice (93), the addition or removal of a trivial MPO to some other MPO symmetry can be made trivial:

$$
\begin{aligned}
&= \delta_{k,n}\delta_{b,c} \;,\\
&= \delta_{j,n}\delta_{a,c} \;,
\end{aligned}
\tag{107}
$$

where we use the following states to terminate or create a vacuum line:

$$
= \;\; = \delta_{n,1}\delta_{B,A}\sqrt{d_A}.
\tag{108}
$$

The existence of the particular states in Eq. (108) also allow us to define explicitly the left and right evaluation and coevaluation morphisms at the level of the MPO symmetries:

$$
\tag{109}
$$

$$
\tag{110}
$$

Using these morphisms we can write the snake identities in the form

$$
\tag{111}
$$

while tracing becomes

$$A \left( \begin{array}{c} m \\ \bar{a} \\ \alpha \\ a \\ n \end{array} \right) B = A \left( \begin{array}{c} m \\ a \\ \alpha \\ \bar{a} \\ n \end{array} \right) B = d_a \delta_{mn} \; A \left| \begin{array}{c} n \\ \alpha \\ n \end{array} \right| B . \tag{112}$$

The evaluation and coevaluation morphisms establish an important relation between the left and right-handed MPO symmetry tensors [4, 6]:

$$\begin{array}{c} C \overset{n}{\underset{\alpha}{\Big|}} D \\ j \overset{a}{\longrightarrow} \overset{\bar{a}}{\longrightarrow} \overset{a}{\longleftarrow} k \\ A \underset{m}{\Big|} B \end{array} = \begin{array}{c} C \overset{n}{\underset{\alpha}{\Big|}} D \\ j \overset{a}{\longleftarrow} \overset{\bar{a}}{\longrightarrow} \overset{a}{\longrightarrow} k \\ A \underset{m}{\Big|} B \end{array} = \begin{array}{c} C \overset{n}{\underset{\alpha}{\Big|}} D \\ j \overset{}{\longleftarrow} \overset{a}{\longrightarrow} k \\ A \underset{m}{\Big|} B \end{array} . \tag{113}$$

Using this property, we can prove more general forms of the pulling-through equation. In particular we have

$$\Ybar{a} = \Ybar{a} = \Ybar{a} , \tag{114}$$

which gives the relation (14) depicted in the main text.

# D  Examples of bimodule categories

## D.1  The case $\mathcal{C} = \mathcal{M} = \mathcal{D}$

We can always take $\mathcal{M} = \mathcal{C}$, such that the map $\mathcal{C} \times \mathcal{M} \to \mathcal{M}$ is simply the map $\mathcal{C} \times \mathcal{C} \to \mathcal{C}$. In this case, also $\mathcal{D} = \mathcal{C}$ and all the $F$ symbols coincide. The definitions in Eq. (13) then coincide with [4].

## D.2  Finite groups

Take $\mathcal{C} = \text{Vec}_G^\omega$, the category of $G$-graded vector spaces. The simple objects of this category are one-dimensional vector spaces each graded with a different group element $g \in G$; we will identify these simple objects with the group elements themselves and write $g \otimes h = gh$. Using the shorthand $g_1 g_2 = g_{12}$, the associator $^0F$ can be written as

$$\left( {}^0F^{g_1,g_2,g_3}_{g_{123}} \right)^{g_{23},11}_{g_{12},11} \equiv \omega(g_1, g_2, g_3) \tag{115}$$

and the pentagon equation ($P_0$) becomes

$$\omega(g_{12}, g_3, g_4)\omega(g_1, g_2, g_{34}) = \omega(g_1, g_2, g_3)\omega(g_1, g_{23}, g_4)\omega(g_2, g_3, g_4). \tag{116}$$

This equation is known as the 3-cocycle condition, and its solutions are 3-cocycles that are classified by the third cohomology group $\text{H}^3(G, U(1))$. To define MPO tensors that form a representation of the group $G$, we must choose a module category $\mathcal{M}$ over $\text{Vec}_G^\omega$. These have been classified in [59] and can be formulated as follows. Take $H \subset G$ some subgroup of $G$; the

set of simple objects of $\mathcal{M}$ are then the different cosets $gH$ in the set of left cosets $G/H$. The map $\otimes : G \times G/H \to G/H$ is given by

$$g_1 \otimes g_2 H = g_{12}H. \tag{117}$$

If we choose a full set of representatives $m_1, \ldots, m_n$ ($n$ is called the index of $H$ in $G$) for $G/H$ one can readily verify that the associativity condition

$$N_{g_1 g_2}^{g_3} N_{g_3 m_1}^{m_3} = N_{g_1 m_2}^{m_3} N_{g_2 m_1}^{m_2} \tag{118}$$

holds. If we interpret $N_{g_1 m_1}^{m_2} = (N_{g_1})_{m_1}^{m_2}$ as a matrix from $m_1$ to $m_2$ then this equation means that $N_{g_1}$ is a representation of $G$. In group theory, this is known as the induced representation $\mathrm{Ind}_H^G$ of the trivial representation of the subgroup $H$. For a given group $G$ and a 3-cocycle $\omega \in \mathrm{H}^3(G, U(1))$, the pentagon equation ($P_1$) imposes a constraint on the possible choice of subgroup $H$. This can be understood as follows: if we restrict the elements of $G$ to a subgroup $H$ and the elements of $G/H$ to $H$ (with representative $e$), the associator $^1F$ becomes

$$\left(^1 F_e^{h_1 h_2 e}\right)_{h_{12}, 11}^{e, 11} \equiv \psi(h_1, h_2), \tag{119}$$

since $h \otimes H = hH = H$. Using this in pentagon equation ($P_2$), we have

$$\omega(h_1, h_2, h_3) = \frac{\psi(h_1, h_{23}) \psi(h_2, h_3)}{\psi(h_1, h_2) \psi(h_{12}, h_3)} = d\psi, \tag{120}$$

which implies that $\omega$ restricted to the subgroup $H$, denoted as $\omega\mid_H$, can be written as a coboundary and therefore must be trivial. We next treat two particular choices of the subgroup $H$ and construct the invertible bimodules.

## $H = \mathbb{Z}_1$

We can always take $H$ to be the trivial group for any $\omega$. In this case, $G/H = G$ and therefore $\mathcal{M} = \mathcal{C}$, which was already treated in section (D.1).

## $H = G$

This choice can only be made when $\omega$ is trivial. We then have $G/G = \mathbb{Z}_1$, which contains only one element $e$. This means the associator $^1F$ must be of the form

$$\left(^1 F_e^{g_1 g_2 e}\right)_{g_{12}, 11}^{e, 11} \equiv \psi(g_1, g_2) \tag{121}$$

as discussed above. Using this in pentagon equation ($P_2$), we find that $^2F$ must satisfy

$$\left(^2 F_e^{g_{12} e \alpha}\right)_{e, 1 i_3}^{e, i_1 1} = \sum_{i_2} \left(^2 F_e^{g_1 e \alpha}\right)_{e, 1 i_1}^{e, i_2 1} \left(^2 F_e^{g_2 e \alpha}\right)_{e, 1 i_2}^{e, i_3 1} \tag{122}$$

which, writing $\left(^2 F_e^{g_1 e \alpha}\right)_{e, 1 i_1}^{e, i_2 1} \equiv D^\alpha(g_1)_{i_1}^{i_2}$ as matrices becomes

$$D^\alpha(g_{12}) = D^\alpha(g_1) D^\alpha(g_2), \tag{123}$$

meaning the matrices $D^\alpha$ form a representation of $G$. These representations are labeled by $\alpha \in \mathcal{D}$, so we find $\mathcal{D} = \mathrm{Rep}(G)$, and the fact that $\alpha$ label simple objects means that $D^\alpha$ are irreducible representations. Writing $\left(^3 F_e^{e \alpha_1 \alpha_2}\right)_{e, i_1 i_2}^{\alpha_3, k s i_3} \equiv C_{i_1 i_2 i_3, k}^{\alpha_1 \alpha_2 \alpha_3}$ in equation ($P_3$), we find that it should satisfy

$$\sum_{j_3} C_{i_1 i_2 j_3, k}^{\alpha_1 \alpha_2 \alpha_3} D^{\alpha_3}(g)_{j_3}^{i_3} = \sum_{j_1, j_2} D^{\alpha_1}(g)_{i_1}^{j_1} D^{\alpha_2}(g)_{i_2}^{j_2} C_{j_1 j_2 i_3, k}^{\alpha_1 \alpha_2 \alpha_3}. \tag{124}$$

This equation implies that $C$ is the intertwiner between the tensor product of two irreducible representations $D^{\alpha_1}(g) \otimes D^{\alpha_2}(g)$ and the irreducible representation $D^{\alpha_3}(g)$, which means that $C$ is a Clebsch-Gordan coefficient. In this notation, $k$ labels the degeneracies, i.e. the different ways in which $\alpha_1$ and $\alpha_2$ can fuse to $\alpha_3$. Equation ($P_4$) implies that this fusing process must be associative:

$$\sum_{j_6} C^{\alpha_1 \alpha_6 \alpha_4}_{i_1 j_6 i_4, m} C^{\alpha_2 \alpha_3 \alpha_6}_{i_1 i_2 j_6, n} = \sum_{\alpha_5, lk j_5} \left( {}^4F^{\alpha_1 \alpha_2 \alpha_3}_{\alpha_4} \right)^{\alpha_6, nm}_{\alpha_5, kl} C^{\alpha_1 \alpha_2 \alpha_5}_{i_1 i_2 j_5, k} C^{\alpha_5 \alpha_3 \alpha_4}_{j_5 i_3 i_4, l}, \tag{125}$$

with associator ${}^4F$. In group theory this object is known as the Racah W-coefficient; they are related to the $6j$ symbols by a phase, and they are a solution of pentagon equation ($P_5$).

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
