# Peer review of "Matrix product operator symmetries and intertwiners in string-nets with domain walls"

_SciPost Physics, doi:SciPost Phys. 10, 053 (2021)_

## Round 2 · Referee Report · Anonymous (Referee 1) · 2020-11-16

Report

The authors examine the structure of MPO symmetries in PEPS states. In particular, they identify that the complete set of equations obeyed by the PEPS tensors, virtual MPO symmetry tensors, and the associated fusion tensors, as the set of pentagon equations obeyed by two fusion categories $\mathcal{C},\mathcal{D}$, and a $(\mathcal{C},\mathcal{D})$ bimodule category. The PEPS tensor itself is realized by the `module' F symbols ${}^3!F$, allowing the generalization of previous string-net PEPS to the full Morita class. Additionally, explicit virtual MPO intertwiners between different string-nets are examined.

Finally, the authors show how these PEPS representations, realized by general invertible bimodules, correspond to TFTs, and therefore place them on a firm mathematical footing.

As the authors state, realizing PEPS via distinct module categories in this way may be useful for studying phase transitions out of the string-net phase. In particular, how the underlying microscopic model impacts this behavior. It is the framework for this work that I see as the main contribution of this work.

I find this paper both interesting, and generally well written, and recommend publication. I have some recommended changes:

  • The first mention of the bimodule associators ${}^1!F-{}^3!F$ on page 8 occurs without their definition (apart from a brief reference to the appendix). Since these equations (13) are already rather complicated, this made the section difficult to follow on first reading. At the very least, I'd recommend a more direct reference to the relevant sections of the appendices, including how to read the `triple line notation'.

  • I found the discussion of the TFT boundaries rather confusing. This is not my area of expertise, and I struggled to follow why the two boundaries $\Sigma\times{0}$ and $\Sigma\times{1}$ are treated so differently. Should I think of the choice of a physical boundary as `using up' that end of the manifold?

---

## Round 2 · Referee Report · Anonymous (Referee 2) · 2021-1-24

Report

The authors use bimodule category to formulate the basic concepts of topologically ordered phases using tensor networks. It is shown in detail how the consistency conditions arising from non-local matrix-product operator (MPO) symmetries are equivalent to pentagon equations. These bimodule categories furthermore allow to construct MPO “intertwiners” between different projected entangled pair states (PEPS) representations of the same string-net. This result provides a generalization of virtual gauge transformations between PEPS that describe the same state.

The manuscript is on an exciting topic and overall clearly written. The “dictionary” between the formulation in terms of topological fields theories and tensor networks will be useful for a practitioners in the field. As pointed out by the referees, the framework could be very useful for simulations of error-correcting codes based on string-net models.

I thus find the manuscript interesting and sufficiently relevant for publication in SciPost Physics after addressing the minor comments below.

Requested changes

  • I feel that the existing literature on topological phase transitions should be cited more prominently. The concept of anyon condensation has for example been pioneered in F. A. Bais and J. K. Slingerland Phys. Rev. B 79, 045316.

  • Overall, it might make it more clear when emphasizing which parts are reformulations of existing relations and what are new results that emerge from the formulation in terms of tensor networks.

  • Given that the “pull through condition” is used extensively, it might be helpful to introduce it in some more detail right in the beginning.

---

## Round 3 · Author Response

We thank the referees for their reading of the manuscript and their positive reports. To address the comments of the first referee:

The first mention of the bimodule associators on page 8 occurs without their definition (apart from a brief reference to the appendix). Since these equations (13) are already rather complicated, this made the section difficult to follow on first reading. At the very least, I'd recommend a more direct reference to the relevant sections of the appendices, including how to read the `triple line notation'.

We have added additional information on these associators before they are used in equations (13), and provided a direct reference to the relevant section in the appendix. A more detailed explanation and their precise definition would require the background of fusion, module and biomdule categories, which we feel would obscure the fact that these objects appear naturally in the tensor network description.

I found the discussion of the TFT boundaries rather confusing. This is not my area of expertise, and I struggled to follow why the two boundaries are treated so differently. Should I think of the choice of a physical boundary as `using up' that end of the manifold?

We have added an additional paragraph explaining what would happen if we consider both boundary conditions to be either gluing or physical boundaries. The physical boundaries have no degrees of freedom on them and can be thought of as boundary conditions; in this sense, they indeed `use up' a particular part of the manifold.

Adressing the comments of the second referee:

I feel that the existing literature on topological phase transitions should be cited more prominently. The concept of anyon condensation has for example been pioneered in F. A. Bais and J. K. Slingerland Phys. Rev. B 79, 045316.

We have added references on this particular topic.

Overall, it might make it more clear when emphasizing which parts are reformulations of existing relations and what are new results that emerge from the formulation in terms of tensor networks.

The tensor network perspective, including the various equations that in the end correspond to pentagon equations of a bimodule category, has been previously described and various properties of these systems (i.e. topological entanglement entropy) have been studied in the tensor network language before. We think the main contribution of this work is the realization that the MPO symmetries and the string-net model do not have to be described by a single fusion category, but rather that there is additional flexibility in this description and that a complete description of MPO symmetries requires the consideration of invertible bimodule categories, which is mentioned in the introduction. This observation provides insight into the nature of PEPS itself, but we also expect it to provide a useful framework for considering e.g. phase transitions. Other new results include a PEPS description of boundaries and domain walls in string-net models, which requires the flexibility of these more general representations, as well as a generalized interpretation in terms of Turaev-Viro state sum models with boundary.

Given that the “pull through condition” is used extensively, it might be helpful to introduce it in some more detail right in the beginning.

We have added extra motivation for this pulling through condition, and why it is a natural feature of models with topological order.

---

## Round 3 · List of Changes

- Various spelling/grammar mistakes corrected
- Additional explanation on the pulling-through conditions in Section 2
- Additional reference to the appendix regarding the bimodule associators used in Eqs (13)
- Corrected a mistake in Eq (27)
- Additional explanation on the difference between gluing and physical boundary in Section 5, after Eq (51)

---

## Editorial Decision

published